**Registered report**

# Deliberately ignoring inequality to avoid rejecting unfair offers

Konstantin Offer [1,2,3] ✉, Dorothee Mischkowski [4,5], Zoe Rahwan [1] & Christoph Engel [4]

Why do people punish experienced unfairness if it induces costs for both the punisher and punished person(s) without any direct material benefits for the punisher? Economic theories of fairness propose that punishers experience disutility from disadvantageous inequality and punish in order to establish equality in outcomes. We tested these theories in a modified Ultimatum Game (*N* = 1370) by examining whether people avoid the urge to reject unfair offers, and thereby punish the proposer, by deliberately blinding themselves to unfairness. We found that 53% of participants deliberately ignored whether they had received an unfair offer. Among these participants, only 6% of offers were rejected. As expected, participants who actively sought information rejected significantly more unfair offers (39%). Averaging these rejection rates to 21%, no significant difference to the rejection rate by participants who were directly informed about unfairness was found, contrary to our hypothesis. We interpret these findings as evidence for sorting behavior: People who punish experienced unfairness seek information about it, while those who do not punish deliberately ignore it.

**Protocol registration**

The Stage 1 protocol for this Registered Report was accepted in principle on 13 October 2023. The protocol, as accepted by the journal, can be found at https://doi.org/10.6084/m9.figshare.24559132.v1.

Costly punishment occurs when individuals inflict harm on others, at a cost to themselves. A person intending to maximize their profit would obviously not do so. Yet, this behavior even occurs in one-shot interactions[1,2] and is fundamental for the promotion of cooperation between genetically unrelated individuals[3–5], serving important evolutionary functions[6–8]. Explanations for costly punishment typically focus on strong reciprocity, enforcements of fairness norms, and social preferences for equitable outcomes[9]. However, unfair behavior can only be reciprocated, fairness norms enforced, and social preferences for equitable outcomes followed if people know about the unfairness. A growing body of evidence indicates that people often deliberately remain ignorant. They intentionally avoid information that might threaten their self-esteem or lead to materially disadvantageous outcomes[10]. Here, we studied *deliberate ignorance*, defined as the conscious choice not to seek or use information[11], as a strategic device to avoid being confronted with disadvantageous inequality that might provoke an urge to punish. We experimentally introduced uncertainty about inequality in a canonical economic game, the Ultimatum Game (UG), and empirically tested the prediction that people avoid costly punishment by deliberately ignoring free information on unfair treatment.

There is a growing body of evidence originating from psychology[10–13], economics[14–16], and neuroscience[17] that deliberate ignorance is not an exception to the rule, but rather frequent and widespread[13,14,18,19]. While information avoidance may be unconscious[15,20], deliberate ignorance requires conscious choice. This can occur for strategic reasons[21], drawing on distinct psychological motives (e.g., gaining bargaining advantages[22], eschewing responsibility[23], or avoiding liability[24]). For example, people may exploit "moral wiggle room" by choosing not to know how their choices affect others[23,25,26] or the natural environment[27]. In such cases, deliberate ignorance allows individuals to maintain a positive (self-)image while still benefiting from the consequences of their self-serving decisions[28].

We contribute to the growing body of evidence on deliberate ignorance in the context of economic games by addressing an important and hitherto unanswered question about the relationship between deliberate ignorance and costly punishment. Previous experimental evidence shows that deliberate ignorance can be a shelter from punishment[29,30]: When people intentionally choose not to know whether their decisions create situations in which they are better off at the cost of others, the probability of being punished is reduced. Further, third parties refrain from punishment when they can do so without

[1]Max Planck Institute for Human Development, Center for Adaptive Rationality (ARC), Lentzeallee 94, 14195 Berlin, Germany. [2]Max Planck School of Cognition, Stephanstrasse 1a, Leipzig, Germany. [3]Humboldt-Universität zu Berlin, Department of Psychology, Berlin, Germany. [4]Max Planck Institute for Research on Collective Goods, Bonn, Germany. [5]Leiden University, Leiden, The Netherlands. ✉e-mail: offer@mpib-berlin.mpg.de

revealing their preferences[31], and may ignore inequalities to avoid inducing costs[32,33]. In contrast, evidence is lacking on the side of people who have been wronged, leaving the question open as to whether individuals deliberately ignore being treated unfairly to avoid punishing others. Real-world examples span from intimate relationships (where a partner might choose to overlook infidelity to preserve the relationship) to broader societal conflicts (where nations might deliberately overlook hidden provocations and attacks to resist the impulse of retaliation). The present study attempted to close this gap by introducing uncertainty about inequality in the UG, and we expected individuals to avoid the urge to punish by deliberately ignoring free information on disadvantageous inequality.

The UG is a canonical economic game commonly used to study costly punishment. In the standard UG, an anonymous proposer receives a fixed amount of money and makes an offer to a responder regarding the split of the money. The responder knows how much money the proposer has received and accepts or rejects the offer. If the offer is accepted, the proposed split is implemented. If the offer is rejected, both players get nothing. Unfair offers (i.e., offers below 50% of the endowment) are often rejected, and the probability of rejection increases the more the split is asymmetric[34–37]. Since costly punishment is without any direct material benefit for responders in the UG, and even creates opportunity costs, it has also been described as altruistic punishment[1,3]. At the same time, there is evidence that, for some, UG punishment is spiteful rather than altruistic[38], that costly punishment can be conducted by both fair-minded and unfair-minded punishers[39], and that altruistic punishment is not more prevalent in real-world altruists than in controls[40]. This suggests that UG punishment is social, but not necessarily conducted out of prosocial or altruistic motives. Following this evidence, we refer to UG punishment as costly punishment in the remainder of the article.

Costly punishment is commonly explained with economic theories of fairness, or self-centered inequality aversion, which incorporate social preferences for equitable outcomes into standard economic theories relying solely on self-interest[41,42]. One of the most widely-applied economic theories of fairness that can explain costly punishment in the UG is the Fehr and Schmidt model[41]. The Fehr and Schmidt model suggests that the utility of some responders is not just dependent on their absolute payoff, but also on their payoff in relation to the proposer's payoff. If the responder's disutility from disadvantageous inequality exceeds their utility from the monetary payoff, then a responder is expected to reject an ultimatum offer. That is, rejections in the UG depend on the responder's aversion to disadvantageous inequality and the size of the offered share (for a formal description of the UG predictions, see Supplementary Note 1).

One implicit assumption in the Fehr and Schmidt model is that the size of the share is known to responders. An unknown size of the share comes with an uncertain (dis-)utility, since the responder cannot assess the degree of unfairness. For example, if a responder has been offered 1$, but does not know whether the proposer has split 2$ or 10$ (and does not know with which probability the proposer's endowment is high or low), then the

responder is unable to infer their (dis-)utility derived from possible inequality. If the proposer has split 2$ (evenly), it is utility-maximizing to accept the offer, since it consists of a fair split without any disutility. In contrast, if the proposer has split 10$, responders with high aversion to inequality would not accept the offer if they knew about the inequality, since their disutility from disadvantageous inequality would exceed their utility from the monetary payoff.

Introducing uncertainty about disadvantageous inequality (e.g., whether 2$ or 10$ have been split), with a corresponding option to freely seek information, might shed light on the role of deliberate ignorance in the context of punishment. Here, a desire for complete information and a benefit of the doubt might compete, where the latter can spare the responder costly punishment. In this situation, it is only rational to seek information on the amount of money the proposer has received, as long as disutility from incomplete information exceeds the utility from the monetary payoff. By not seeking information on the proposer's endowment, responders who would have rejected the offer under certainty are no longer able to say whether a rejection is justified or not. Based on their ignorance, these inequality-averse responders can exploit the benefit of the doubt and avoid the urge to punish, leaving both parties better off in material terms. However, if the same responders chose to seek information, discovering that 10$ had been split, then rejection would be the dominant choice, leaving both parties with a payoff of zero.

We theorized an extension of the Fehr and Schmidt model for costly punishment in the UG by introducing exogenous uncertainty about disadvantageous inequality (see Fig. 1). Uncertainty is defined as an informational state in which a responder receives an offer, does not know how much money the proposer has received and with what probability the endowment is high or low, and can seek information on the proposer's endowment at no extra cost. Deliberate ignorance is defined as a responder's conscious choice not to seek information on the amount of money the proposer has received. Punishment is defined as the costly rejection of an ultimatum offer, and inequality is a split of the money in favor of the proposer. In line with classic models for costly punishment[41,42], under certainty the model simplifies to an expected positive effect of inequality on punishment, since ignorance can only occur under uncertainty.

Introducing uncertainty about disadvantageous inequality led to three hypotheses. First, we expected a lower probability of punishment by responders who initially did not know the size of the share (i.e., under uncertainty), as compared to responders under certainty. Second, we expected the effect of uncertainty on punishment to be larger for high, as compared to moderate, inequality, as deliberate ignorance could be beneficial at high levels of inequality for regulating emotions[11,43]. For example, the probability of punishment by responders who received 1$ and did not know whether 2$ or 10$ had been split was expected to be affected more strongly by uncertainty than the probability of punishment by responders who received 1$ and did not know whether 2$ or 3$ had been split. Finally, we expected a lower probability of punishment for ignorant than for non-ignorant responders in the case of uncertainty, as we predicted that inequality-averse responders would deliberately ignore free information on inequality to avoid punishment. That is, we expected deliberate ignorance to reduce the probability of punishment under uncertainty.

The remainder of the introduction is organized by the three research questions that we wanted to answer. For each question, we discuss competing hypotheses as well as interpretations for data patterns that are in line with and contrary to our predictions. A summary of our questions, hypotheses, sampling plans, analysis plans, and interpretations for different data patterns is presented in Table 1.

Our first research question (RQ1) focused on the main effect of uncertainty on punishment. Is uncertainty about inequality exploited to avoid punishment, even if information can be sought at no extra cost? We predicted a lower probability of punishment in the case of uncertainty (H1A), since inequality-averse responders have an interest in ignoring unfairness, as long as the monetary benefit of accepting the offer is greater than the disutility from not knowing.

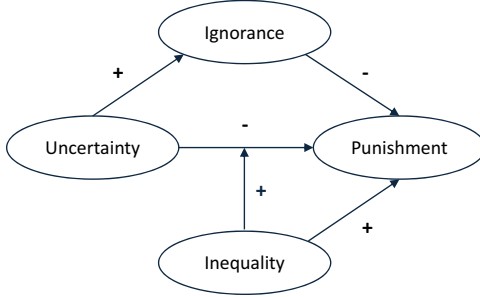

**Fig. 1 | A simple model of punishment under uncertainty in the UG.** We expected a lower probability of punishment under uncertainty (H1A; hypothesis for RQ1). Inequality was expected to moderate this negative relationship between uncertainty and punishment (H2A; hypothesis for RQ2). Since uncertainty allows for ignorance, we expected that ignorance reduces the probability of punishment (H3A; hypothesis for RQ3).

**Table 1 | Design Table**

| Question | Hypothesis | Sampling plan (e.g., power analysis) | Analysis Plan | Interpretation given to different outcomes |
|---|---|---|---|---|
| **RQ1:** Does uncertainty about inequality affect punishment (i.e., rejection rates in the UG)? | **H10:** No difference in punishment between certainty and uncertainty. **H1A:** Lower probability of punishment under uncertainty (vs. certainty). **H1B:** Higher probability of punishment under uncertainty (vs. certainty). | **Power Analysis:** Two-sided, two-sample t-test with the pwr package[56]. **Assumptions:** (1) *Cohen's D* = −0.39 (2) *Tails* = 2 (3) *Significance Level* = 0.05 (4) *A Priori Power* = 0.9 **Result:** $n_1 = 283$. | **Analysis Plan RQ1:** Regression of punishment (y) as the dependent variable on uncertainty (x) as the independent variable in a linear probability model. | **R10:** Reinforcement of a fairness norm based on altruistic punishment. **R1A:** Exploitation of exogenous uncertainty to avoid costly punishment. **R1B:** Need for consistency in active compared to passive information acquisition. |
| **RQ2:** Does the level of inequality moderate the effect of uncertainty on punishment? | **H20:** No interaction between uncertainty and inequality on punishment. **H2A:** Interaction effect between uncertainty and inequality in that the effect of uncertainty is *smaller* for moderate, as compared to high, inequality. **H2B:** Interaction effect between uncertainty and inequality in that the effect of uncertainty is *larger* for moderate, as compared to high, inequality. | **Power Analysis:** Interaction effect in a 2×2 factorial design under variance heterogeneity[57]. **Assumptions:** (1) $\mu_{cmod} = 0.28$; $\mu_{chigh} = 0.68$ $\mu_{uncmod} = 0.18$; $\mu_{unchigh} = 0.41$ (2) $\sigma_{cmod} = 0.45$; $\sigma_{chigh} = 0.47$; $\sigma_{uncmod} = 0.38$; $\sigma_{unchigh} = 0.49$ (3) *Significance Level* = 0.05 (4) *Sample Size Ratio* = 1:1:1:1 (5) *A Priori Power* = 0.9 **Result:** $n_2 = 1{,}200$. | **Analysis Plan RQ2:** Regression of punishment (y) as the dependent variable on uncertainty (x), as the independent variable and inequality (z), as the moderating variable, plus the interaction term between x and z in a linear probability model. | **R20:** Punishment in line with classic economics of information and economic theories of fairness. **R2A:** Greater exploitation of exogenous uncertainty at higher levels of inequality. **R2B:** Eagerness to detect and punish high (vs. moderate) inequality. |
| **RQ3:** Given uncertainty about inequality, does ignorance lead to reduced punishment? | **H30:** No difference in punishment under uncertainty between ignorant and non-ignorant responders. **H3A:** Lower probability of punishment for ignorant than for non-ignorant responders. **H3B:** Higher probability of punishment for ignorant than for non-ignorant responders. | **Power Analysis:** Two-sided, two-sample t-test with the pwr package[56]. **Assumptions:** (1) *Cohen's D*=−1.05 (2) *Tails* = 2 (3) *Regression for X* =1 (4) *Significance Level* = 0.05 (5) *A Priori Power* = 0.9 **Result:** $n_3 = 80$. | **Analysis Plan RQ3:** Regression of punishment (y) as the dependent variable on ignorance (v) as the endogenous predictor variable in a linear probability model for all subjects in uncertainty treatments (x = 1). | **R30:** Punishment in line with classic economics of information and economic theories of fairness. **R3A:** Self-selection into ignorance for avoiding costly punishment under uncertainty. **R3B:** Punishment based on distrust and suspicion in line with earlier work on spitefulness. |

Summary of the questions, hypotheses, sampling plans, analysis plans, and interpretations for data patterns in line with and contrary to our predictions for empirically testing the theorized effects in our model for punishment under uncertainty in the UG.

Two data patterns contrary to our prediction were possible. On the one hand, there could be no association between uncertainty and punishment (H10). Two arguments might support the null hypothesis. First, if responders want to reinforce a fairness norm fostering cooperation through costly punishment[1,3,4], then there should be no reason for responders to avoid costly punishment in the case of exogenous uncertainty. Second, classic economics of information[44] predict that individuals should seek information if potentially beneficial information comes at no extra cost. If an individual holds social preferences, information about relative performance is beneficial. Once free information is sought, there should be punishment in line with classic economic theories of fairness, resulting in no difference in punishment between certainty and uncertainty.

On the other hand, there could also be a higher probability of punishment under uncertainty (H1B) due to a need for consistency. That is, if a sufficiently large proportion of responders chooses to resolve uncertainty by seeking information on whether they have been treated unfairly, they may also want to punish the experienced unfairness in order to be consistent in their behavior. One may also wonder whether there is (an extension of) a sunk cost effect: If participants have already overcome their hesitation and retrieved the information, they may feel compelled to act upon it. As a result, individuals may be more likely to punish unfairness if they actively sort themselves into information environments where they encounter it. Data contrary to our prediction would, consequently, be interpreted as evidence for differences in punishment between active and passive information acquisition, in line with earlier studies on sorting behavior[28,45,46].

Based on these arguments, we derived the following hypotheses:

**H10:** There will be no difference in punishment between certainty and uncertainty.
**H1A:** There will be a lower probability of punishment under uncertainty (vs. certainty).
**H1B:** There will be a higher probability of punishment under uncertainty (vs. certainty).

Our second research question (RQ2) addressed the interaction between uncertainty and inequality. Is the effect of uncertainty on punishment conditional on the potential degree of inequality (i.e., on the size of the highest possible endowment)? We expected that the effect of uncertainty on punishment would be smaller for moderate, as compared to high, potential inequality (H2A). There were two reasons for this prediction. First, as the offered share decreases, more rejections may be expected, as the probability for rejection and the size of the offered share are negatively related[34–37]. As a result, uncertainty if inequality is potentially high could induce responders to exploit uncertainty. Second, higher levels of inequality lead to stronger negative affect[43], which could be regulated by the conscious choice not to know[11].

Similar to RQ1, two data patterns could be in conflict with our prediction. On the one hand, there could be no interaction between uncertainty and inequality (H20). There were two arguments in favor of H20. First, if all responders behaved in line with the Fehr and Schmidt model, there should be no difference in punishment between certainty and uncertainty, and no interaction between uncertainty and inequality. Second, higher inequality comes, by definition, with a greater difference between fair and unfair offers. This might induce more information search for some, possibly counter-acting less information search due to higher inequality for others – resulting in a null effect on the population level due to inter-individual differences. On the other hand, the effect of uncertainty on punishment could also be larger for moderate than for high inequality (H2B). Here, the reasoning might be that, if the proposer is only a little bit better off, it is not worth knowing. But given the risk of severe exploitation, one may choose to know. We thus expected the effect to be driven by a desire to know about high inequality, leading to higher punishment.

**H20:** There will be no interaction between uncertainty and inequality on punishment.

**H2A**: There will be an interaction effect between uncertainty and inequality in that the effect of uncertainty is smaller for moderate, as compared to high, inequality.
**H2B**: There will be an interaction effect between uncertainty and inequality in that the effect of uncertainty is larger for moderate, as compared to high, inequality.

Our third research question (RQ3) focused on the effect of ignorance on punishment under uncertainty. Can ignorance predict differences in punishment under uncertainty? We predicted a lower probability of punishment for ignorant than for non-ignorant responders (H3A). We made this prediction for two reasons. First, there is evidence that responders accept significantly lower offers when they cannot seek information on how much money is being divided[47,48]. Second, responders who want to avoid costly punishment are expected to choose ignorance strategically, since ignorance about inequality can help them to avoid the urge to punish an unfair proposer, and may preserve self-esteem, serving as an excuse for not punishing.

Contrary to our prediction for RQ3, there could be either no effect (H30) or a positive effect (H3B) of ignorance on punishment. The main argument in favor of H30 was similar to the deduction of the null hypotheses above: If responders behaved in line with classic economics of information[44] and rejected offers in line with economic theories of fairness, then there should be no ignorance; and without variance in ignorance, there could not be covariation with punishment. In contrast, there could very well be a higher probability of punishment for ignorant as compared to non-ignorant responders: The former may experience regret and suspicion after making irreversible choices not to seek information on how much money the proposers have allocated to themselves. Consequently, data contrary to our prediction would be interpreted as evidence for distrust-based rejections of ultimatum offers in line with earlier studies on punishment and spitefulness[49,50].

**H30**: There will be no difference in punishment under uncertainty between ignorant and non-ignorant responders.
**H3A**: There will be a lower probability of punishment for ignorant than for non-ignorant responders.
**H3B**: There will be a higher probability of punishment for ignorant than for non-ignorant responders.
In sum, the aim of the present study was to test the prediction that people avoid costly punishment by deliberately ignoring free information about possible disadvantageous inequality. In doing so, we aimed at contributing to the literature on deliberate ignorance by testing its generalizability to punishment behavior. While the intentions behind deliberate ignorance of disadvantageous inequality may be selfish, the outcome could be Pareto-optimal, since both parties end up with higher payoffs if one party avoids costly punishment.

## Methods
### Ethics information
The study was approved under the ethical regulations of the Max Planck Institute for Research on Collective Goods in Bonn, Germany. The study was incentivized, and no deception was used. Informed consent was obtained from all participants. Participant compensation was at least 6£/8$ per hour (in line with Prolific's pricing policy), plus a possible bonus payment ranging between 2¢ and 90¢.

### Design
To answer the three research questions, we used a modified mini-ultimatum game[36,51] in a 2 × 2 factorial design. In our experimental design, we defined uncertainty as the independent variable (x) and inequality as the moderating variable (z). Ignorance (v) was an endogenous predictor variable, and punishment (y) the dependent variable. All four variables were binary variables, and an overview of the 2×2 factorial design is provided in Fig. 2.

**Fig. 2 | 2x2 factorial design.** The design consisted of four experimental treatments. There was one independent variable and one moderating variable. The independent variable was uncertainty, and the moderating variable inequality. Inequality could be moderate (defined as a 70:30 split) or high (defined as a 90:10 split). In the two uncertainty treatments, responders chose between free information and ignorance. Ignorance was the endogenous predictor variable. All responders chose between rejection and acceptance. Punishment was the dependent variable, which was operationalized as the costly rejection of an unfair ultimatum offer.

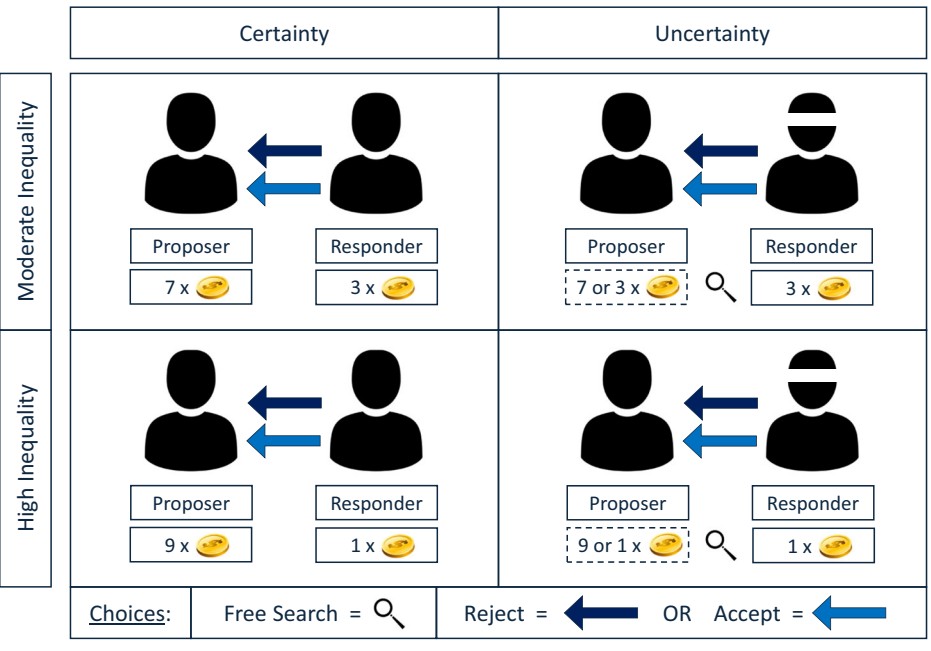

The independent variable was uncertainty, defined as an informational state. This state could be either certain ($x = 0$) or uncertain ($x = 1$). Under certainty, the responder was automatically informed about the size of the pie, and hence directly saw whether the offer was equal or unequal. Under uncertainty, the responder initially only saw how much money they would get if they accepted the offer. The responder did not know how much money the proposer had received and was provided with information on the two possible amounts (i.e., high or low endowment) that the proposer could have received, but did not learn with which probability the endowment was high or low. The responder could then seek information on the amount at no extra cost. For example, a responder may have been offered 10¢ with the information that the proposer had either received 20¢ or 100¢. The responder then decided whether to retrieve information on the proposer's endowment before accepting or rejecting the offer.

The moderator variable was inequality which could be moderate ($z = 0$) or high ($z = 1$). Moderate inequality was realized by a 70:30 split in favor of the proposer, whereas high inequality was operationalized as a 90:10 split. Proposers always chose between an equal (i.e., 50:50) and an unequal split, which depended on the inequality condition. Actually, the endowment was 100¢ across all treatments, with the exception of a small share of participants in uncertainty treatments who received endowments of 20¢ and 60¢, to avoid deception.

The endogenous predictor variable was ignorance, which we operationalized as a responder's conscious choice not to seek information within the uncertainty condition. Moreover, we elicited descriptive beliefs of ignorant responders by asking whether they expected the offer to be equal or unequal. Beliefs were measured after responders had decided whether to accept or reject the ultimatum offer (but before the uncertainty was resolved) in order not to bias the measurement of the dependent variable.

The dependent variable was punishment, which was defined as the costly rejection of an ultimatum offer. Data on punishment was collected by asking all responders whether they wanted to accept ($y = 0$) or reject ($y = 1$) an unfair ultimatum offer.

We started our investigation whether deliberate ignorance influences punishment by relying on a one-shot setting (i.e., without any repeated interactions). We did so mainly to study causal links in the absence of feedback. However, we generally expected that the basic insights from our study also apply to repeated interactions, where individuals may deliberately ignore that they might have been treated unfairly.

Our experimental procedure consisted of a seven-step Qualtrics-based online experiment (see Fig. 3) applying a variant of the strategy method[52] with a sample of $N = 1370$ (for power analyses, see sampling plan below). First, participants based in the US were recruited via Prolific and completed a consent form. Participants were then randomly assigned to one out of four between-subjects treatments, varying in uncertainty (yes vs. no) and inequality (moderate vs. high). In all treatments, participants received UG instructions in the role of the responder and completed a comprehension check to confirm that they understand the game. Second, participants were informed that they had been offered 30¢ (in treatments with moderate inequality) and 10¢ (in treatments with high inequality). Third, participants in uncertainty treatments were informed that proposers had received either 100¢ or 60¢ (in treatments with moderate inequality) and 100¢ or 20¢ (in treatments with high inequality), before being given the option to retrieve the endowment by clicking a button, based on the "hidden information" condition by Dana et al.[23] as implemented by Grossman[53]. Recall that the endowment was held constant at 100¢ across all treatments, with the exception of a small and randomly selected share of participants with endowments of 60¢ and 20¢, thereby ensuring that no deception was applied; participants were told that the endowment might be "either 100¢ or 20¢" (but were not told with which probability the endowment was high or low). Participants in certainty treatments and participants who decided to seek information were informed about the endowment. Correspondingly, participants who decided not to know were not informed. Fourth, subjects were asked whether they wanted to accept or reject the offer. Fifth, subjects in uncertainty treatments had the option to state their reasons for (not) seeking information, and subjects in uncertainty treatments who had not sought information were asked to indicate their beliefs about the amount of money the proposer had split.

Even though steps one to five would have sufficed to answer the three research questions, two more steps were implemented to ensure that there was no deception for (even) proposer offers. That is why, in a sixth step, all participants were informed about the endowment and asked whether they wanted to accept or reject an even offer. Finally, participants switched roles and chose, as proposers, whether they offered 30¢ or 50¢ (in treatments with moderate inequality) and 10¢ or 50¢ (in treatments with high inequality). Importantly, the order of all seven steps was fixed to ensure that participants did not erroneously infer the recipient's share from the choice as a proposer. Note that fixing the order of questions did not create any order effects for responses to unfair offers,

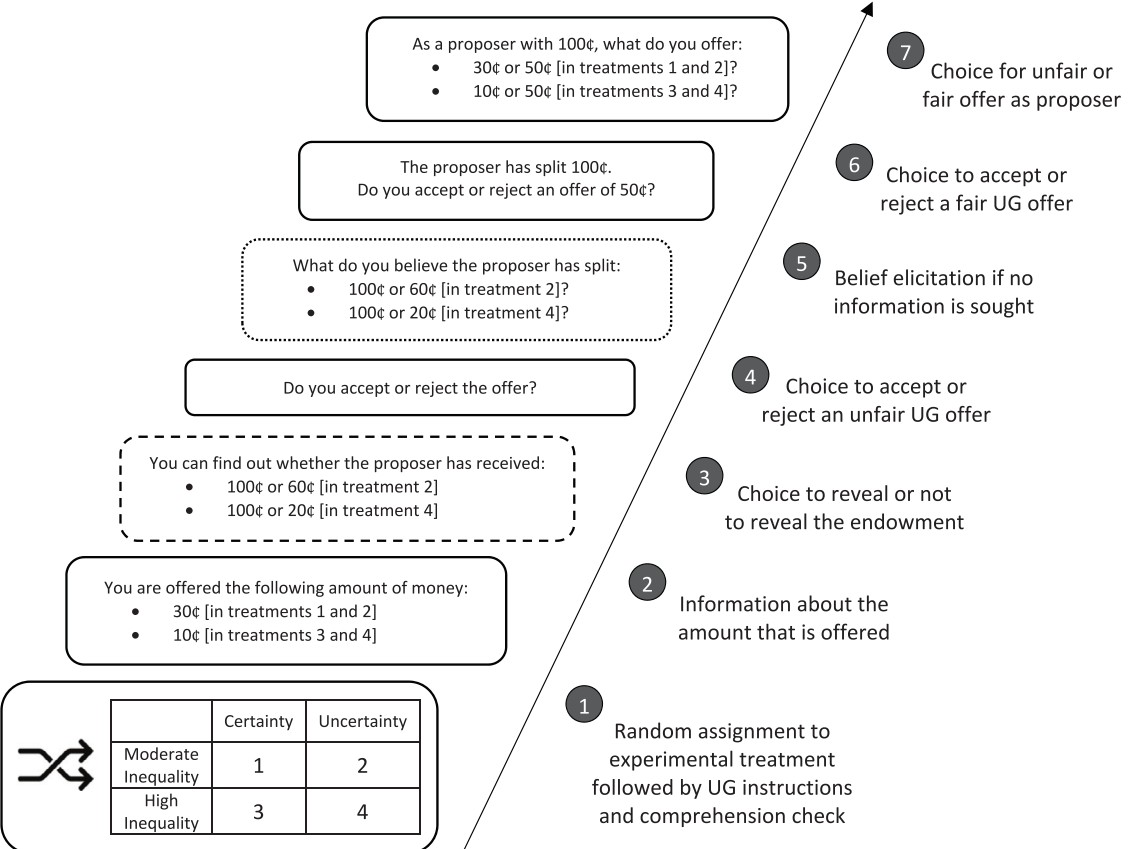

**Fig. 3 | Experimental procedure.** Note that UG refers to Ultimatum Game and that the option to seek information (here: dashed line) only occurred in uncertainty treatments (i.e., treatments 2 and 4). Beliefs were elicited (here: dotted line) after offers were accepted or rejected and only in cases where participants chose not to seek information.

since responses to unfair offers were always elicited first. Feedback was withheld until the end of the experiment, when all participants were thanked and debriefed.

The matching of participants and calculation of payoffs took place once all data had been collected. To do so, half of the participants in each treatment were randomly assigned to the proposer role. All remaining participants were responders, and each responder was randomly matched with one proposer in their treatment. Given the proposers' choices in step seven, either fair or unfair offers were made. This step ensured that all offers were real and without deception. Based on the responders' choices in steps four and six, offers were either accepted or rejected. If an offer was accepted, the proposed split was implemented in the form of a bonus payment. If an offer was rejected, no bonus was paid.

## Sampling plan

For each of the three research questions, we conducted an a priori power analysis. We based all of them on $1 - \beta = 0.9$ to reach a high and economically achievable statistical power with $\alpha = 0.05$. To test our hypotheses most conservatively, we based the sampling plan on the largest calculated sample size.

We derived our effect size estimates from historic UG rejection rates and a recent meta-analytic review of deliberate ignorance. In particular, we analyzed rejection rates from 17 UG studies with varying levels of inequality[34], resulting in expected mean probabilities of rejection for high and moderate inequality under certainty of 68% and 28%, respectively. Further, based on a recent meta-analytic review of 22 studies on deliberate ignorance[54], we expected a probability of ignorance of 0.40 for moderate inequality and 0.44 for high inequality. Based on our pilot data (see Supplementary Notes 2 and 3), we expected that the probability of rejection under ignorance would not be larger than 0.10. The resulting probabilities of rejection under moderate inequality and uncertainty and high inequality and uncertainty were 0.18 and 0.41, respectively. Standard deviations for the four probabilities of rejection were calculated as $\sigma_{cmod} = 0.45$, $\sigma_{chigh} = 0.47$, $\sigma_{uncmod} = 0.38$, and $\sigma_{unchigh} = 0.49$ given the Bernoulli distributions.

The first hypothesis test was based on a bivariate linear regression. We preferred a linear probability model over logit or probit for consistency with the model for the second hypothesis test focusing on an interaction effect[55]. To calculate the required sample size, we used the pwr.t.test function in R (version 4.3.1) for a two-sided, two-sample t-test[56]. The required sample size consisted of $n_1 = 283$ participants. The second power analysis focused on the identification of an interaction effect in a linear probability model. We calculated the required sample size with a specialized R program for examining interaction effects in factorial designs under variance heterogeneity[57]. This yielded a required sample size of $n_2 = 1200$. The third hypothesis was tested by a bivariate linear regression on half of the sample (i.e., for participants where $x = 1$), since ignorance could only occur in uncertainty treatments. In the same way as for the first power analysis, we used the pwr.t.test function for a two-sided, two-sample t-test. A sample of $n_3 = 80$ was required. The sampling plan was most conservatively based on $N = 1,230$ (i.e., 300 subjects per treatment plus 30 subjects with endowments of 20¢ and 60¢ to avoid deception) to ensure that there is power of $1 - \beta = 0.9$ even for the second hypothesis.

Given our power analyses, we saw the possibility of "over-powering" our analyses for RQ1 and RQ3. To avoid interpreting very small differences that are statistically significant but neither theoretically nor practically relevant, we committed ourselves to only interpreting effect sizes of 10% or more of our derived effect sizes (see Table 1) as meaningful. That is, we would interpret effect sizes of $d_1 < 0.04$ and $d_3 < 0.11$ for RQ1 and RQ3, respectively, as too small to be of theoretical or practical relevance.

There were three exclusion criteria. First, we only compared choices in UGs with endowments of 100¢ to ensure comparability across treatments. Second, we excluded participants who self-reported careless participation[58]. In particular, we did not include choices by participants who answered "no" to the question: "Honestly, should we use your data in the analysis of our study?". Third, we asked three comprehension questions to assess whether participants understood the game. For their data to be included in the analysis, participants had to answer all three questions correctly within two attempts. That is, all participants who still gave at least one wrong answer in their second attempt were excluded from our analysis. To account for potential selection effects (e.g., based on differences in general cognitive ability or conscientiousness), we conducted a robustness check in which we analyzed whether results differed when including non-understanding individuals (see Supplementary Note 4). Since the results did not differ when including non-understanding individuals, results from the full sample are reported.

## Analysis plan

All statistical analyses and data manipulations were performed using the R programming language, and regression models were built with the R "stats" package (version 4.3.1). For the significance level, we used $\alpha = 0.05$ for RQ1, RQ2, and RQ3 to test our preregistered hypotheses.

The aim of RQ1 was to examine whether uncertainty affects punishment in the UG. In order to test our hypotheses for RQ1, we conducted a bivariate linear regression in which we predicted the probability of punishment ($y$) by uncertainty ($x$) for individuals $i = 1, \ldots, n$, with $\varepsilon_i$ being the error term:

$$y_i = \beta_{10} + \beta_{11} x_i + \varepsilon_i \tag{1}$$

We relied on linear probability models for all of our tests, as the interaction effect herein corresponds to the marginal effect of the interaction term, unlike interaction effects in logit models[55]. To answer RQ2 (which asked whether the effect of uncertainty on punishment is moderated by inequality), we regressed punishment ($y$) on uncertainty ($x$), inequality ($z$) and the interaction term:

$$y_i = \beta_{20} + \beta_{21} x_i + \beta_{22} z_i + \beta_{23} x_i z_i + \varepsilon_i \tag{2}$$

To examine whether ignorance predicts punishment under uncertainty (RQ3), we conducted a bivariate linear regression predicting the probability of punishment ($y$) by ignorance ($v$) for all participants within uncertainty treatments ($x = 1$):

$$y_i = \beta_{30} + \beta_{31} v_i + \varepsilon_i \tag{3}$$

The requirements for the three regression models were verified in preceding analyses. In particular, we assessed whether predictions of less than 0.05 or more than 0.95 were made by our linear probability models. If such predictions occurred, we would report additional logit models next to our respective linear probability models to support the robustness of our estimates. Before we interpreted $\beta_{11}$, $\beta_{23}$, and $\beta_{31}$, we assessed the overall model fit of our three regression models in terms of explained variance on the basis of $\alpha = 0.05$. If the overall model fit of a regression model was not statistically significant, we would not interpret any regression coefficients and discard the model altogether. We did not plan any further post hoc inclusions of control variables on the basis of our preregistered hypotheses.

## Pilot data

We conducted two pilot studies with participants from Prolific ($n_1 = 165$, $n_2 = 164$) to assess the feasibility of our paradigm (see Supplementary Notes 2 and 3). The objectives of our pilot studies were to (I) ensure that our task can detect a positive effect of inequality on punishment and (II) provide initial information on the proportion of individuals who choose not to seek information. In our first pilot study, we implemented the design as for the main study, specified above. The pilot study detected a positive effect of inequality on punishment and revealed a ceiling effect in information search. The second pilot study allowed us to address the ceiling effect. In our second pilot study, we examined information search under moderate inequality for varying instructions and costs for seeking information. The second pilot study revealed neither a floor nor a ceiling effect in the search for free information. Moreover, it provided preliminary evidence that ignorance has a positive effect on punishment – in line with H3A.

The first pilot study provided evidence for a positive effect of inequality on punishment. Participants in treatments with high inequality had a significantly higher probability to reject unfair offers than participants in treatments with moderate inequality (inequality = 0.38, $t(143) = 5.055$, $p < 0.001$, 95% CI [0.235, 0.525]; see Supplementary Fig. 2.1). This finding was in line with our assumed effect size in the power analysis and rejection rates reported in the literature[34]. Further, the first pilot study revealed a ceiling effect in information search: 95% of participants in moderate and high inequality conditions decided to seek information. We hypothesized that this ceiling effect resulted from deviations from study designs previously used in the literature. To further examine this expectation, we conducted a second pilot study.

The second pilot study examined information search for varying instructions and costs for seeking information (see Supplementary Note 3). The pilot study consisted of four conditions, all of which had moderate inequality and uncertainty. The first condition was designed as a control condition employing the same instructions as in the first pilot study. In particular, participants in this condition were not told how the proposer's endowment had been determined, whether the other person would be informed about their information seeking or not, and whether the interaction would be anonymous or not. Information on the proposer's endowment could be sought by clicking a button labelled "reveal other person's money" – making "no reveal" the default choice as in previous studies[23,28,29,59]. The second condition employed instructions as described by Grossman[53]. More specifically, participants in this condition were told that the endowment had been randomly determined by a computer, that the other person would not be informed about their information seeking, and that the interaction would be anonymous. Information on the proposer's endowment could be sought by selecting one of two buttons labelled "Proceed" and "Reveal version", with the "Proceed" button preselected – making "no reveal" the default choice, as in previous studies and condition one. Conditions three and four were identical to condition two with the only difference that they introduced additional costs of 10¢ and 20¢ for seeking information, respectively.

The mean probabilities of ignorance for conditions one, two, three, and four were 25%, 64%, 95%, and 100%, respectively. These findings suggested that costs for seeking information could be expected to lead to a floor effect in information search, and that the instructions employed in condition two (where participants were informed about how the proposer's endowments had been determined, whether the other person would be informed about their information seeking or not, and whether the interaction would be anonymous or not) could be expected to neither lead to a floor nor a ceiling effect in information search. Based on these findings, instructions from condition two were used in the main study to employ a study design for which neither floor nor ceiling effects would be expected.

While the detection of a positive effect of ignorance on punishment was not the primary objective of our pilot studies, our second pilot study nonetheless provided preliminary evidence for it. In particular, among the 31% of participants that chose to seek information, 33% rejected an unfair offer – broadly in line with historic UG rejection rates[34]. In contrast, among the 69% of our participants who chose not to know whether they had been treated unfairly, only 3% rejected the offer.

## Reporting summary

Further information on research design is available in the Nature Portfolio Reporting Summary linked to this article.

**Table 2 | Demographic Information**

| Variable | Frequency (*n*) | Proportion (%) |
|---|---|---|
| Gender | | |
| Men | 733 | 51 |
| Women | 671 | 47 |
| Other (e.g., non-binary) | 14 | 1 |
| Prefer not to say | 12 | 1 |
| Education | | |
| High school degree | 383 | 27 |
| Associate's degree | 183 | 13 |
| Bachelor's degree | 587 | 41 |
| Master's degree or PhD | 232 | 16 |
| Other (e.g., trade certificate) | 27 | 2 |
| Prefer not to say | 18 | 1 |
| Ethnicity | | |
| Caucasian or white | 943 | 66 |
| African or African American | 159 | 11 |
| Asian or Asian American | 157 | 11 |
| Hispanic, Latino, or Latina | 90 | 6 |
| Native American or Indigenous | 9 | 1 |
| Multiracial or Mixed | 51 | 4 |
| Other (e.g., Russian Jewish) | 7 | < 1 |
| Prefer not to say | 14 | 1 |

The characteristics of our survey respondents are based on answers to the following three questions: 1) "What is your gender?" 2) "What is your highest level of education completed?" and 3) "Which of the following best describes your ethnicity?". We recruited 1430 US participants via Prolific based on our power analyses.

## Results
### Sample description
In pilot study one, we had an exclusion rate of 12% based on our predefined exclusion criteria. In pilot study two, the exclusion rate was 16%. To reach our predefined sample size of $N = 1230$ with an expected exclusion rate of 14%, we recruited $N = 1430$ US participants via Prolific. The mean age of our participants was 41 years (SD = 13). Table 2 displays demographic information on our participants.

In total, 99 of our recruited participants did not fulfill the three predefined inclusion criteria: 53 participants played a UG with an endowment of 60 or 20 cents to avoid deception, 7 participants stated that their data should not be included in the data analysis due to non-seriousness in participation, and 39 participants failed to answer all three comprehension questions correctly within two attempts. As our results were robust to whether or not the participants passed or failed the comprehension test (see Supplementary Note 4), we consequently only excluded 60 participants from our data analysis (i.e., those with a different endowment and those who reported unserious participation), in keeping with our data analysis plan. The sample sizes for our four treatments with certainty and moderate inequality, uncertainty and moderate inequality, certainty and high inequality, and uncertainty and high inequality were 358, 351, 314, and 347, respectively. We report observations for these 1370 participants in the remainder of the article.

### Data quality and manipulation checks
Analogous to our two pilot studies, we had two manipulation checks to assess our data quality. In particular, we wanted to (I) assess whether our task could detect a positive effect of inequality on punishment and (II)

ensure that we faced neither a floor nor a ceiling effect in the proportion of individuals who chose not to seek information (i.e., v = 1).

To assess whether our task can detect the classic inequality effect, we regressed punishment on inequality. As in our pilot study, we found that inequality significantly predicted punishment ($\beta_{01} = 0.17$, t(1368) = 7.987, $p < 0.001$, 95% CI [0.129, 0.211]). The rejection rate under moderate inequality was 12% (t(1368) = 8.063, $p < 0.001$, 95% CI [0.091, 0.149]). Further, 52.6% (95% CI [0.488, 0.563]) of participants ignored inequality. Specifically, the ignorance rates under moderate and high inequality were 56.7% (95% CI [0.513, 0.619]) and 48.4% (95% CI [0.431, 0.538]), respectively. Hence, we detected the classic inequality effect and neither faced a floor nor a ceiling effect in the proportion of individuals who chose not to know.

In accordance with our preregistration, we verified whether our linear probability models made predictions below 0.05 or above 0.95. As all rejection rates were within the interval of 0.05 and 0.95, we report results from our linear probability models, without additional reporting of logit models.

### Uncertainty and punishment
To recap, we tested our hypotheses for RQ1 by predicting the probability of punishment ($y$) by uncertainty ($x$). The overall linear probability model (1) was not significant (F(1, 1368) = 1.613, $R^2 = 0.001$, SE = 0.40, $p = 0.204$). The rejection rates under certainty and uncertainty were 18.6% and 21.3%, respectively. This difference was not significant ($\beta_{11} = 0.03$, t(1368) = 1.27, $p = 0.204$, 95% CI [−0.013, 0.073]). Hence, we could not reject H10. To answer RQ2, we regressed punishment ($y$) on uncertainty ($x$), inequality ($z$) and the interaction term. The overall linear probability model (2) was significant (F(3, 1366) = 21.78, Adj. $R^2 = 0.044$, SE = 0.39, $p < 0.001$) due to the significant main effect of inequality on punishment ($\beta_{22} = 0.18$, t(1366) = 6.044, $p < 0.001$, 95% CI [0.121, 0.239]). Yet, we did not find the expected interaction between uncertainty and inequality ($\beta_{23} = −0.03$, t(1366) = −0.675, $p = 0.500$, 95% CI [−0.112, 0.052]), such that we could neither reject H20. Figure 4 displays the effects of uncertainty and inequality on punishment.

### Ignorance and punishment
To examine whether ignorance predicts punishment under uncertainty (RQ3), we predicted the probability of punishment ($y$) by ignorance ($v$) for all participants within uncertainty treatments ($x = 1$). The overall regression model was significant (F(1, 696) = 133.8, $R^2 = 0.161$, SE = 0.38, $p < 0.001$). In line with our expectation (H3A), we found that ignorance significantly predicted punishment in linear probability model (3) ($\beta_{31} = −0.33$, t(696) = −11.57, $p < 0.001$, 95% CI [−0.385, −0.275]). Specifically, among the 52.6% of participants who chose not to know whether inequality was present, only 5.7% of unfair offers were rejected, while 38.7% of the participants who chose to know rejected unfair offers (t(696) = 18.72, $p < 0.001$, 95% CI [0.342, 0.432]).

### Exploratory analyses
Three informational states were possible in our experiment. First, all participants in certainty treatments were directly informed of any inequality. The information acquisition by these participants was passive. Second, 52.6% of all participants in uncertainty treatments chose not to know. These participants were deliberately ignorant. Third, the remaining 47.4% of participants in uncertainty treatments consciously chose to know. Their information search was active.

Three comparisons were possible: One could compare (1) deliberately ignorant and information-seeking responders, (2) deliberately ignorant and directly informed responders, and (3) directly informed and information-seeking responders. The first comparison was a comparison between active states, while the second and third comparisons were comparisons between active and passive states. Our analysis plan only focused on the first comparison: In line with our expectation (H3A), we found differences in the punishment rates by ignorant and information-seeking responders. Yet, we

**Fig. 4 | Effects of uncertainty and inequality on punishment.** In line with previous studies, we found that inequality significantly predicted punishment: Under moderate inequality, responders rejected 10.1% and 13.7% of unfair offers given certainty ($n_{MC} = 358$) and uncertainty ($n_{MU} = 351$), respectively, while 28.3% and 29.1% of unfair offers were rejected for high inequality given certainty ($n_{HC} = 314$) and uncertainty ($n_{HU} = 347$). Yet, no effect of uncertainty on punishment and no interaction between uncertainty and inequality was found. Error bars represent 95% confidence intervals.

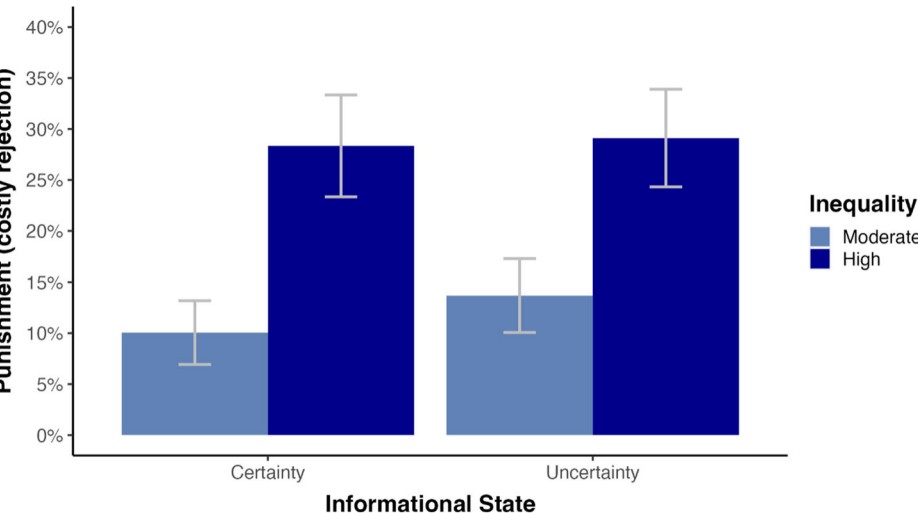

**Fig. 5 | Effects of informational state on punishment.** In line with our hypothesis (H3A), we found a significant difference in the probability of punishment between deliberately ignorant and information seeking responders. Follow-up analyses revealed significant differences between all three informational states (***$p < 0.001$). Responders who chose not to know whether they had received an unfair offer ($n_1 = 367$) rejected only 5.7% of unfair offers, while responders who chose to know ($n_3 = 331$) rejected 38.7% of unfair offers. Under certainty ($n_2 = 672$; a state where responders were directly informed about inequality), 18.6% of unfair offers were rejected. Error bars represent 95% confidence intervals.

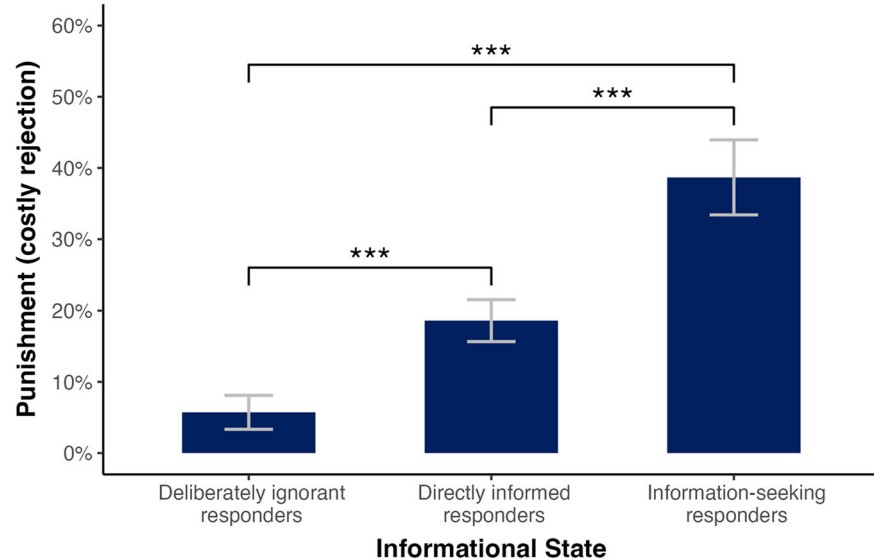

did not preregister any of the other two comparisons. To analyze all cases and close this gap, we ran two additional analyses, using a Bonferroni-adjusted $\alpha_{adj.} = 0.05/3$ to account for multiple comparisons (for regression models, see Supplementary Note 5).

Our first additional analysis compared the mean probability of punishment for deliberately ignorant ($s_{d1} = 0$) and directly informed ($s_{d1} = 1$) responders. Regressing punishment on $s_{d1}$, we found a significant difference in punishment ($\beta_{51} = 0.129$, t(1037) = 5.796, $p < 0.001$, 95% CI [0.086, 0.172]). The mean probability of punishment by deliberately ignorant responders was 5.7% (t(1037) = 3.202, $p = 0.001$, 95% CI [0.022, 0.092]). Analogously, we found a significant difference in punishment between directly informed and information-seeking responders ($\beta_{61} = 0.201$, t(1001) = 7.043, $p < 0.001$, 95% CI [0.144, 0.258]) based on a mean probability of punishment of 18.6% by directly informed responders (t(1001) = 11.363, $p < 0.001$, 95% CI [0.155, 0.217]). The mean probability of punishment by information-seeking responders was 38.7% (t(1001) = 16.580, $p < 0.001$, 95% CI [0.342, 0.432]). Taken together, our two additional analyses revealed significant differences in the mean probability of punishment between all three informational states (see Fig. 5).

For the case of a failure to reject H20, we pre-committed ourselves to a follow-up analysis to assess whether a null effect on the population level resulted from more information search due to uncertainty aversion by some,

possibly counteracting less information search due to inequality aversion by others (see Peer Review File in Supplementary Information). In particular, we assessed this alternative explanation based on participants' agreement scores ($\omega$) to the statements "I chose to find out as I wanted to know about possible inequality" and "I chose not to find out as I did not want to know about possible inequality" (depending on their choice (not) to seek information). We hypothesized that if the alternative explanation was correct, then seeking information due to uncertainty aversion and not seeking information due to inequality aversion would cancel each other out. That is, the agreement scores ($\omega$) would not predict ignorance ($v$) after controlling for inequality ($z$). In particular, we regressed $v$ on $\omega$ and $z$:

$$v_i = \beta_{40} + \beta_{41}\omega_i + \beta_{42}z_i + \varepsilon_i \qquad (4)$$

We found that the agreement scores significantly predicted ignorance. In particular, higher agreement scores were associated with lower probabilities of ignorance ($\beta_{41} = -0.1$, t(695) = −10.21, $p < 0.001$, 95% CI [−0.120, −0.080]). Hence, we rejected the alternative explanation that more information search due to uncertainty aversion by some possibly counteracted less information due to inequality aversion by others.

There were two possible explanations for the lack of evidence against H10 and H20. First, observed choices in uncertainty treatments could be

**Table 3 | Observed and Predicted Choices for Certainty and Uncertainty Treatments**

| | Certain | | | | Uncertain | | |
|---|---|---|---|---|---|---|---|
| | Observed | Predicted | | | Observed | | |
| | | No reveal | Reveal | Total | No reveal | Reveal | Total |
| Accept | 547 | 333 | 195 | 528 | 346 | 203 | 549 |
| Reject | 125 | 20 | 123 | 143 | 21 | 128 | 149 |

Based on our observations in uncertainty treatments, we predicted the distribution of choices in certainty treatments to assess whether the observed and predicted distributions in certainty treatments were invariant. Predicted "No reveal" and "Reveal" are counterfactual scenarios given the observed choices under uncertainty. The predictions were based on the assumptions that the distributions in certainty and uncertainty treatments were identical and that the effects in uncertainty treatments were exclusively driven by heterogeneity.

driven by heterogeneity among participants. Specifically, participants might respond differently to inequality. This explanation is in line with systematic associations between social value orientation (SVO) – a measure for individuals' preferences toward resource distributions in social situations – and the rejection of unfair UG offers[60–62]. Specifically, prosocial individuals might want to know about unfairness to sanction it. In contrast, individuals with a proself orientation may be reluctant to impose costs associated with information search and subsequent punishment, thus preferring ignorance. As a second explanation, uncertainty treatments could have provided a richer context to which participants reacted differently. In particular, perceived proposer intentions could have differed between treatments with certainty and uncertainty. In treatments with certainty, proposers who made unfair offers were in all cases openly selfish. In treatments with uncertainty, some responders may have, in contrast, expected proposers to hide behind opaqueness, possibly hoping for ignorance. Some responders may have disliked such strategic proposer considerations, which were only possible in uncertainty treatments. Evidence on the relevance of perceived intentions[63] and UG proposer features[64] might support this second explanation, focusing on contextual differences between treatments rather heterogeneity among participants.

While we did not have direct measures of the two competing explanations, we could assess whether the observed choices in uncertainty treatments were invariant to the choices in certainty treatments. If we could not exclude invariance between treatments, the first, personality-based explanation would be indirectly supported. In particular, we could predict reactions of participants in certainty treatments from reactions of participants in uncertainty treatments for counterfactual scenarios in which no information had been disclosed on whether the proposer had received a large endowment. These predictions were based on the assumptions that, in uncertainty treatments, we learned something about the type distribution in the population, and that this distribution was identical in certainty and uncertainty treatments (i.e., the observed effects in uncertainty treatments were exclusively driven by heterogeneity among participants). Table 3 displays the observed and predicted distribution of choices in certainty treatments based on our observations in uncertainty treatments.

If the failure to reject H10 and H20 was the result of exogenous heterogeneity, observed and predicted choices in the certainty treatments should be indistinguishable. Running a chi square test on the observed and total predicted choices in certainty treatments, we could indeed not rule out that choices were taken from the same distribution ($\chi^2 = 1.379$, df = 1, $p = 0.240$). While indirect, these findings suggested that the failure to reject H10 and H20 might have been driven by heterogeneity among participants rather than contextual differences. Hence, both this finding and our results on the agreement scores and differential punishment by informational states (see Fig. 5) point towards an explanation of punishment under uncertainty based on individual differences (i.e., social preferences). Taken together, we interpret the observed choices as an indication for sorting behavior: Some reveal unfairness and subsequently punish, whereas others deliberately ignore it without inflicting costs.

## Discussion

The study was designed to assess whether uncertainty affects the probability of punishment (RQ1), whether inequality moderates this effect (RQ2), and whether conscious choices not to know can predict the probability of punishment (RQ3). We found a significantly higher probability of punishment by non-ignorant than by ignorant responders (H3A) but we could not reject the null hypotheses of no difference in punishment between certainty and uncertainty (H10) and no moderation by inequality (H20). We interpret these results as evidence for sorting behavior in that people who punish experienced unfairness actively seek information about unfair ultimatum offers, whereas those who decide not to punish tend to deliberately ignore such information.

In line with our hypothesis (H3A), we found a strong negative relationship between deliberate ignorance and costly punishment. We had based this hypothesis on two reasons. First, responders accept significantly lower offers when they cannot seek information on how much money is being divided[47,48]. Exogenously introduced ignorance reduces the probability of punishment. Second, responders may use ignorance strategically to avoid punishment and preserve their self-esteem. We provide evidence that previous results for exogenous ignorance may generalize to endogenous ignorance: More than half of our participants chose not to know that they had been treated unfairly, and among these participants only 6% of offers were rejected. In contrast, the rejection rate for information-seeking responders was 39%. Based on the taxonomy for deliberate ignorance[11], we interpret this difference in terms of emotion regulation and as a strategic device for the preservation of self-esteem. Specifically, choosing not to know that one has been treated unfairly can be used strategically for anticipating and countering possible anger, resentment, or distrust. Maintaining positive self-esteem, individuals may accept possibly unfair offers at lower emotional costs and at reduced levels of cognitive dissonance.

Contrary to our expectations, we could not reject two null hypotheses in that there was no difference in punishment between certainty and uncertainty (H1A) and no interaction between uncertainty and inequality on punishment (H2A). Importantly, these findings seem to be, at least in part, driven by sorting behavior into different informational states, which we had already anticipated prior to our data collection (see H1B). Specifically, we had theorized that individuals who actively sort themselves into informational environments that reveal potential unfairness may be more likely to punish unfairness. We stated that data contrary to our prediction would be interpreted as evidence for differences in punishment between active and passive information acquisition, in line with earlier studies on sorting behavior[28,45,46]. Two exploratory follow-up analyses allowed us to provide empirical support for this sorting-based explanation. In particular, we found that responders in uncertainty treatments divided almost evenly into responders who sought (47.4%) and responders who ignored (52.6%) information. Responders who sought information rejected 39% of unfair offers, while responders who ignored information rejected only 6% of unfair offers. Both responses significantly differed from the 19% rejection rate in certainty treatments. Yet, taken together, they formed an average rejection rate of 21% in uncertainty treatments, not significantly different from the 19% rejection rate in certainty treatments. A predefined follow-up analysis ruled out that the failure to reject H20 could be explained by competing motives for seeking information due to uncertainty aversion and not seeking information due to inequality aversion. Rather, we interpret costly punishment as the reinforcement of a fairness norm in line with economic theories of fairness (for our pre-registered interpretations, see R10 and R20 in Table 1). Based on previous findings on the relationship between SVO and UG punishment[60–62], it is possible that participants who are willing to punish experienced unfairness seek information about it, while those who are unwilling to punish may choose to deliberately ignore it. Yet, future research is needed to examine the role of SVO in the relationship between deliberate ignorance and UG punishment.

There are important differences between our findings on sorting behavior and costly punishment, and earlier findings on sorting behavior and (charitable) giving. In both types of sorting behavior, individuals move in or out of economic environments given their social preferences. In the case of giving behavior, sorting holds the potential to reduce individuals' sharing with others. For example, charitable giving has been reduced by 28% to 42% in a door-to-door fund-raiser when households were informed about the time of solicitation with an upfront flyer, accompanied with the option to check a box marked "Do Not Disturb"[45]. Such behavior has been interpreted with reference to two types of motivation: Some individuals truly like to give (e.g., due to warm glow or altruism)[65], while others prefer to avoid giving but do not like to say "no" (e.g., due to social pressure)[66]. Contrary to giving behavior, we did not find evidence for these two types of motivation when it comes to costly punishment. Regardless of the informational environment (i.e., certainty vs. uncertainty), participants rejected around 20% of unfair offers. If there had been a second type of motivation (i.e., individuals who prefer to avoid punishment, and mainly punish because they feel pressured to), a lower probability of punishment would have been expected under uncertainty. Yet, this is not what we found. While findings on exploitations of "moral wiggle room"[23,25,26] and avoidances of giving based on sorting behavior[45,46] can pose challenges for economic theories of fairness[41,42], our findings on costly punishment align well with existing theories.

### Limitations

There are several limitations to our study. First, we worked with relatively low stakes in a canonical economic game, the UG. While inequality was moderate (70:30) and high (90:10) in our treatments, the costs for punishment were relatively small (i.e., 30¢ and 10¢ in unfair offers). It is possible that participants would have avoided punishment if the cost of punishment had been larger or if punishment had involved non-monetary costs (e.g., as is the case when confronting a partner about infidelity). Second, we studied one-shot interactions without direct measures of individual differences (e.g., social preferences). Deliberate ignorance in repeated interactions will require further theorizing and more complex experimental designs. Future research could extend our work by examining repeated interactions and directly measuring social preferences for equitable outcomes to assess their associations with deliberate ignorance. Finally, all of our subjects were US participants from Prolific. While their demographics cover a broad range within the US (see Table 2), limitations to the generalizability beyond the US apply. In particular, variations in sorting behavior can be expected across societies, given substantial variations in costly punishment across populations[7]. Future research could unravel the extent to which sorting behavior into states of knowing and not knowing differs across populations, possibly at varying stake sizes and for repeated interactions.

### Conclusions

Our study advances the literature on costly punishment by allowing for sorting into different informational states. There are two important findings. First, more than half of our participants did not want to know that they had been treated unfairly. Second, participants who chose not to know that they had been treated unfairly punished significantly less (a rejection rate of 6%) compared to participants who chose to know that they had been treated unfairly (a rejection rate of 39%). Ignorance can be used in preventive ways in that individuals may shield themselves from knowledge that is potentially costly, in both financial and psychological terms. Importantly, many situations outside of the laboratory occur neither under perfect certainty nor under uncertainty which cannot be resolved. Instead, individuals often have a choice to find out about unfairness, allowing for sorting behavior into different informational states. For example, a spouse may have the option to confront their partner about infidelity, and a nation state can collect evidence about a rival's hostile behavior. Our data suggests that a substantial proportion of people may choose not to know about possible unfairness, and that self-sorting behavior may be an important explanation for real-world punishment behavior. Individuals who may not want to punish, may not

want to know, as not knowing allows them to exploit the benefit of the doubt. The consequence is knowledge gaps between those who are willing and unwilling to punish: Some people seek information about experienced unfairness, while others avoid it.

### Data availability
All data for this study have been made publicly available in an anonymized form at the OSF.

### Code availability
All analyses pipelines, power analyses scripts, and study materials for this study have been made publicly available at the OSF.

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

## Acknowledgements
The study is supported by BMBF and Max Planck Society, and the experiment is funded by the regular budget of the Max Planck Institute for Research on Collective Goods, Bonn, Germany. The authors received no other specific funding for this work next to the regular budget.

## Author contributions
K.O. developed the research idea. K.O., D.M., Z.R. and C.E. delineated the hypotheses and designed the study. K.O. conducted the study, analyzed the data, and wrote a first draft of the article, which was jointly revised by K.O., D.M., Z.R. and C.E.

## Funding

## Competing interests
The authors declare no competing interests.
