## [Peer Review File · Communications Psychology]

14th Apr 23

Dear Mr Offer,

Thank you once again for your manuscript, entitled "Avoiding Punishment by Ignoring Unfairness: Responses to Uncertain Inequality in the Ultimatum Game," and for your patience during the peer review process.

Your manuscript has now been evaluated by 3 reviewers, whose comments are included at the end of this letter. Although the reviewers find your protocol to be of interest, they also raise some important concerns. We are very interested in the possibility of proceeding further with your submission in *Communications Psychology*, but would like to consider your response to these concerns in the form of a revised manuscript before we make a decision on in principle acceptance and Stage 2 submission.

To guide the scope of the revisions, the editors discuss the referee reports in detail within the team, we identified key priorities that should be addressed in a revisions.

1) You are introducing a new paradigm (see also reviewer comments on design changes and exclusion criteria below) that is previously untested. Registered Reports receive in-principle acceptance before the research project commences. However, in exceptional cases, Stage 2 protocols are still rejected, because it becomes apparent that the data are unsuitable, or basic manipulation checks fail. This should be avoided this at all costs. For this reason, we ask authors of new paradigms to provide feasibility pilot data. For this purpose, we ask you to run a small scale study in which you test the appropriateness of your exclusion criteria, your attention checks, and the suitability of the data for the planned analysis (for example, do the manipulations appear to have any effect on behaviour, are outcome neutral conditions required for interpretation, do participants show floor/ceiling effects?). This undertaking will also allow you to deposit your analysis pipeline (code) a priori to commencing the actual research.

2) The Reviewers ask for a justification of the expected effect size — this aspect is very important in a RR and needs to be carefully addressed. Editorially, we require an effect size justification not only for the secondary analyses (which are likely overpowered), but also the main targeted effect size. Effect size estimates may either be based on the existing literature, where they need to rest on cumulative evidence from the literature (not single studies as effect sizes in individual studies are likely inflated, especially given historically small samples). In the absence of a body of work to provide an effect size estimate and an appropriate justification for the chosen effect size in relation to the literature, you may also settle for the smallest effect size of theoretical or practical importance. This requires a scientific argument for why smaller effect sizes would not be of interest/relevance. Reviewer#2 has also important suggestions on how to interpret results that will test effects that are expected to be bigger than the smallest effect size of interest. Finally, the best strategy for the power analysis would be to run it on the same statistical model that will be used to test the hypothesis. In the case of linear regressions, it can be implemented with several packages such as SIMR in R for instance.

2) Reviewer #1 raises an important point for the interpretation of the results in terms of a deliberative strategy. The reviewer suggests addressing it on a set of analyses on the belief question data. Please adopt these suggestions or explain why the alternative analyses would not benefit the

design.

3) Reviewer #1 has a concern about the potential confounding factor of individual differences in uncertainty reduction over the interpretation of perceived fairness. Please address this concern in the revision.

4) Reviewer #3 and Reviewer #2 ask for a more detailed analytical plan. Some of the missing information is reported in Table 1 and under the sampling strategy section, but that information needs to appear under the analytical plan section. In particular, in this section it is necessary to specify: (1) what software will be used to run the analysis; (2) what significance level will be used; (3) what package will be used to build the regression model; (4) the entire regression models with their respective error terms; (5) how the requirement for the regression model will be verified and what is the plan if there are deviations from the requirements (e.g., outliers and influential cases). When presenting the analytical plan, please remove the claim that “interaction effects cannot be directly tested in non-linear models”. Researchers routinely misinterpreted parameters in GLMs, due to the model's nonlinear link function, which is not an equivalent statement. While it is correct to point out that in general it's not the best idea to directly interpret logit coefficients, and instead generate model predictions (e.g., probabilities) for interesting combinations of predictor levels, the claim needs to be nuanced.

5) Finally, please address the following editorial question about the design: participants in the uncertain condition (2 and 4) can choose to know the initial offer; it may be reasonable to assume that a fair share of participant will select this option (ideally, your pilot data will show some indication of this). How will participants who choose to know the initial offer be treated in the analysis? Experimentally they are assigned to the uncertain treatment, but after choosing to know, they will effectively be in the same experimental condition as treatments 1 and 3. If this is indeed the case, please explain how it will be addressed in the analysis, if it will imply unequal number of participants in each experimental condition and how is this will be considered in the determination of the sample size.

In sum, we invite you to revise your Stage 1 Registered Report taking into account reviewer and editor comments. Please highlight all changes in the manuscript text file.

* Include a “Response to reviewers” document detailing, point-by-point, how you addressed each referee comment. If no action was taken to address a point, you must provide a compelling argument. This response will be sent back to the reviewers along with the revised manuscript.

* Ensure that you use our template for Stage 1 Registered Reports to prepare your revised manuscript:

https://www.nature.com/documents/NHB_Template_RR_Stage1.docx

Failure to ensure that your revised Stage 1 submission meets our requirements as specified in the

template will result in your submission being returned to you, which will delay its consideration.

* In your cover letter, please include the following information:

--An anticipated timeline for completing the study if your Stage 1 submission is accepted in principle.

--A statement confirming that you agree to share your raw data, any digital study materials, computer code, and laboratory log for all eventually published results.

--A statement confirming that, following Stage 1 in principle acceptance, you agree to register your approved protocol on the Open Science Framework (<https://osf.io/>) or other recognised repository, either publicly or under private embargo, until submission of the Stage 2 manuscript.

--A statement confirming that if you later withdraw your paper, you agree to the Journal publishing a short summary of the pre-registered study under a section Withdrawn Registrations.

[link redacted]

We hope to receive your revised manuscript within four to eight weeks. If you cannot send it within this time, please let us know. We will be happy to consider your revision so long as the report still represents a significant contribution to the literature at that stage.

* **TRANSPARENT PEER REVIEW:** Communications Psychology uses a transparent peer review system. This means that we publish the editorial decision letters including Reviewers' comments to the authors and the author rebuttal letters online as a supplementary peer review file. We publish these records for all accepted manuscripts. However, on author request, confidential information and data can be removed from the published reviewer reports and rebuttal letters prior to publication. If your manuscript has been previously reviewed at another journal, those Reviewers' comments would not form part of the published peer review file.

Communications Psychology is committed to improving transparency in authorship. As part of our efforts in this direction, we are now requesting that all authors identified as 'corresponding author' on published papers create and link their Open Researcher and Contributor Identifier (ORCID) with their account on the Manuscript Tracking System (MTS), prior to acceptance. ORCID helps the scientific community achieve unambiguous attribution of all scholarly contributions. You can create and link your ORCID from the home page of the MTS by clicking on 'Modify my Springer Nature account'. For more information please visit www.springernature.com/orcid.

Sincerely,

Marika Schiffer, on behalf of

Eva R. Pool, PhD

Editorial Board Member

Communications Psychology

orcid.org/0000-0001-5929-1007

REVIEWER EXPERTISE:

All reviewers have expertise in behavioural economics. Reviewer #2 serves as a Registered Reports expert.

REVIEWERS' COMMENTS:

Reviewer #1:

Remarks to the Author:

The authors are planning to run a study in which they add an information-seeking choice to the classical ultimatum game (UG) to test whether people avoid costly punishment by deliberately ignoring information about unfairness. If the authors would plan to test these hypotheses, it would be a very interesting study. However, the hypotheses, planned analyses etc don't seem to answer their main research question or at least not directly.

First of all, the authors are trying to bridge the gap between the fields UG and deliberative ignorance, however, it is not clear why this is important and the implication of their results for understanding human behavior. The authors should better specify, for instance with a real-life example, scenarios in which this conflict (information vs. costly punishment) may arise. A concrete and real-life example would also make the manuscript more accessible to a non-expert audience.

Moreover, it is not clear why it should be deliberative ignorance and not just avoidance of information. Our brain tends to avoid negative information (we can consider hurting someone else or discovering that someone is unfair as negative), but not always this is done deliberately. I think the author should clarify this in the intro. In the task, they are forced to make a decision (seek vs. not seek), but in real-life this might not be the case. People may avoid knowing whether another person has behaved unfairly without even realizing it. I think the question about people's belief the authors have added to the task is a clever strategy to solve this issue. However, they should enter this into their analyses as independent variable to understand whether participants will avoid information because they expect to receive something unfair (see below).

Now, in the planned analyses, if I have understood correctly, the authors are using uncertainty as a proxy for informational states (which in fact it is) and they want to see whether uncertainty influence punishments. That's interesting but I do not think this will directly answer the research question. My understanding is that the authors want to know whether people avoid costly punishments by avoiding information. But how the authors can do so if they are not considering the subjects' information-seeking choices? Off-course, with the uncertain condition they are modulating participants' informational state but the only thing they can conclude with that analysis is that enhancing uncertainty reduces punishments, not that participants are avoiding unfairness information to avoid costly punishments. First, it is possible that participants avoid information just

because they do not feel to reduce their uncertainty, and not that they do want to avoid unfairness (see below how these two quantities are actually correlated in the task). There is an intense literature on different motives underlying human information-seeking decisions the authors can refer to (e.g., Kobayashi et al. 2019, NHB, Sharot & Sunstein, 2020, NHB, Cogliati Dezza et al. 2022, Cognition etc..). Second, Q3 can answer whether ignoring can lead to reduced punishments. However, this doesn't mean that people ignore info to avoid punishments. It just means that not having the information will obviously reduce punishments because people do not know whether the responder was fair or not.

I think to answer the research questions the authors need to test 1) whether people ignore information about unfairness – the predicted variable should be information-avoidance (yes, no) and inequality as the predictor (prediction: more avoidance with higher inequality) ; 2) whether this is done deliberately i.e., subjects report that they avoided information because they expected unfairness (for example, a mediation analysis on the relationship between information avoidance and unfairness might help with that) 3) the authors' Q3 - whether avoiding information leads to reduced punishments; 4) whether that relationship is mediated by people's belief. [If the authors decide to run this latter analysis their belief question should off course be asked before the information-seeking decisions. If the authors do not want to interfere with the other variables of the task, this can be run in an independent task Q3]

The other problem that I see is that in the current version of the task inequality and uncertainty (not the authors' uncertainty but the variance between the two offers) are the same thing. In other words, when participants are presented with treatment 4 they are both more uncertain about the possible offer and might incur in higher inequality compared to treatment 2. This means that the valanced of the information (level of unfairness) and the uncertainty are anticorrelated. This can be problematic because participants may want to avoid information because it may reveal unfairness but they do also have the urge to reduce their uncertainty. Now, people who will be more prone to reduce their uncertainty, they might seek more info. People who will be more prone to avoid unfairness they might avoid that information. This might result in a null effect on a population level, without, however, being true. Which are the strategies the authors are planning to use to solve this issue? Related to this point, I'm not sure I have all the details to evaluate this but it seems to me that the information can be useful for participants to improve their bonus payment. Is it? Let's say that a participant is in treatment 4, if they seek information they can reveal that the proposer has received 100. They receive 50 and they accept the offer. The information has helped the participant to add 50 to their bonus payment. Now, if they didn't have that information, they may reject the offer and therefore loose 50. This is another confound in the task. People will seek information, regardless of uncertainty and unfairness, because that information will help improving their bonus payment. Again, how are the authors planning to deal with this issue?

Lastly, the authors confront seeking (free) information with the cost of punishing others. But this implies that they avoid that information because they are making the reasoning if I obtain that info, I have to inflict cost (to themselves and to the other).. but again this might not be the case. Let's say they are not making that reasoning, the authors may find they always seek information, regardless of the inequality, because there is no cost in seeking that information.

Minor

"Introducing uncertainty about disadvantageous inequality (e.g., whether 2\$ or 10\$ have been split), with a corresponding option to freely seek information, sheds light on the role of deliberate

ignorance in the context of punishment.” I think this is a hypothesis. “sheds”  “might shed”

“Deliberate ignorance is defined as a responder’s conscious and irreversible choice not to seek information on the amount of money the proposer has received.” Why irreversible??

Figure 1 is not clear how uncertainty can enable ignorance. By definition, uncertainty enables seeking information to reduce ignorance. In the context of this behavioral task, if there is uncertainty subjects may decide or not decide to resolve it... but it doesn’t mean that it enables ignorance.

Table 1. How did the authors compute the effect size? It would be useful to have pilot data to estimate the effect size.

Figure 3 should also have the moment in which subjects decide whether to seek or not information as it is right now, it seems that subjects doesn't make an info-seek choice

Reviewer #2:

Remarks to the Author:

The authors register a study about uncertainty and unfairness in an economic-social decision-making task. They aim to examine whether uncertainty about (the amount of) inequality influences their decision to punish inequality, whether this process is itself influenced by the amount of inequality, and whether consciously staying ignorant of the amount of inequality reduces the propensity to punish inequality. For this, they developed an experimental between-subjects design which introduces uncertainty about the initial endowment of the proposer in an ultimatum game and the possibility for the responder to receive information about it.

The research plan seems solid: the hypotheses are well grounded in theory and well explained; the experimental design fits the hypotheses; the power analyses are reasonable and reproducible.

However, there are some specifications/details missing or ambiguous which I list below and would suggest the authors clarify before conducting the study.

1. The large sample required to answer RQ2 will “over-power” the analyses for RQs 1 and 3. This is not a problem in itself, but it implies that even smallest deviations between sample means (RQ1) or from an effect size of 0 (RQ3) will easily become statistically significant. Thus, I would recommend adding an estimate of the necessary effect size to be meaningful in your opinion (for research, for every-day life – however you want to define “meaningful”). This is aimed at thwarting interpreting statistically significant effects, which are so small that they are practically meaningless.

2. The next issue regards the exclusion criteria. For the second exclusion criterion mentioned in the Sampling Plan section, please list the specific criteria explicitly that you will use so that readers do not have to read another article (Meade & Craig, 2012) to know the full set of exclusion criteria. Furthermore, you state to ask participants three comprehension questions after the experiment and exclude all those “who fail to answer all three comprehension questions correctly within two attempts”. Does that mean they have to fail all of the three questions to be excluded, that is, they are not excluded if they correctly answer one or two out of three, or are they excluded if they fail any one of the questions (within two attempts)? Please be as precise as possible in the description of any participant exclusion criteria. Furthermore, I would recommend being strict in exclusions due to possible misunderstanding of the instructions. The used task seems to be a one-shot experiment so that each participant answers the crucial trial just once. I understand the rationale for this due to

possible feedback effects. (This might be specified explicitly in the Methods section, though.) However, this design might make findings more susceptible to spurious effects of the wording of instructions and task understanding than designs with numerous trials per participant.

3. Your chosen alpha level (0.05) is implied in the sampling plan in Table 1, but it would be good to specify it explicitly in the Analysis Plan section of the Methods. Furthermore, please report the specifics of any preceding or accompanying analysis you might have planned in more detail (e.g., evaluating the overall model fit/explained variance of the regression models before interpreting regression coefficients; restrictions put on the regression coefficients/procedure how to handle regression coefficients smaller than 0 or larger than 1 (which might happen with linear probability models); a priori planned or criteria for post hoc inclusion of control variables in the analyses; testing the assumed heterogeneity of variance for analysis of RQ2).

4. I would encourage you to add a few questions for the sake of sample description. As research often relies on WEIRD (Western, educated, industrialized, rich, democratic) samples but online studies offer the chance to reach a more diverse sample than studies on-site at universities, it would be interesting for the interpretation and possible later aggregation of this data if you would report some sample characteristics (e.g., age, sex/gender, education level, ethnicity). [With this, I do not aim to encourage you to add these variables in the inference statistical tests but merely use them descriptively for sample characterization.]

References

Meade, A. W. & Craig, S. B. Identifying careless responses in survey data. *Psychological Methods* 17, 437–455 (2012).

Reviewer #3:

Remarks to the Author:

This registered study entitled “Avoiding Punishment by Ignoring Unfairness: Responses to Uncertain Inequality in the Ultimatum Game” (COMMSPSYCHOL-23-0037-T) plans to investigate the effect of deliberate ignorance on the punishment (i.e., reject) in the Ultimatum Game (UG). The UG is an established task to investigate the unfairness and inequality in distribution situations. Deliberate ignorance is a novel factor that influences the response to an offer (accept or reject) in the UG. Hence, this study will be of interest to researchers in related areas. The background and motivation for this study are clear to me, and the sampling plan (i.e., the number of participants based on the assumed effect size) seems valid. However, methods of data analysis are not well described. Therefore, I would like the authors to respond to my concerns below.

1. Model comparison

How many models can be considered in total, and how do they compare (e.g., Akaike information criteria)?

2. Analysis method

Related to the first concern, what methods do you use in your data analysis? Path analysis? Mediation analysis?

Do the authors use a logit model to fit the punishment behavior?

3. Figure 1

Can the authors indicate which arrow corresponds to which hypothesis? This is a minor point.

COMMUNICATIONS PSYCHOLOGY – STAGE 1 RR

POINT-BY-POINT RESPONSES TO REVIEWS

Avoiding Punishment by Ignoring Unfairness:

Responses to Uncertain Inequality in the Ultimatum Game

Konstantin Offer, Dorothee Mischkowski,

Zoe Rahwan, Christoph Engel

31 August 2023

Reviewer #1 Feedback:

0) Study Summary

The authors are planning to run a study in which they add an information-seeking choice to the classical ultimatum game (UG) to test whether people avoid costly punishment by deliberately ignoring information about unfairness. If the authors would plan to test these hypotheses, it would be a very interesting study. However, the hypotheses, planned analyses etc don't seem to answer their main research question or at least not directly.

Thank you very much for your interest in our study. We hope that our revisions to the manuscript and our point-by-point responses to your review will fully address the important concerns that you raise.

1) Real-life Examples

First of all, the authors are trying to bridge the gap between the fields UG and deliberative ignorance, however, it is not clear why this is important and the implication of their results for understanding human behavior. The authors should better specify, for instance with a real-life example, scenarios in which this conflict (information vs. costly punishment) may arise. A concrete and real-life example would also make the manuscript more accessible to a non-expert audience.

Thank you for highlighting that real-life examples had been missing and that their addition would make our manuscript more accessible to a non-expert audience. We have added the following two specific examples (marked in **bold**) to our introduction in the manuscript (p.3, para. 1) to address this concern: "Real-world examples span from intimate relationships (**where a partner might choose to overlook infidelity to preserve the relationship**) to broader societal conflicts (**where nations might deliberately overlook hidden provocations and attacks to resist the impulse of retaliation**)."

2) Information Avoidance

Moreover, it is not clear why it should be deliberative ignorance and not just avoidance of information. Our brain tends to avoid negative information (we can consider hurting someone else or discovering that someone is unfair as negative), but not always this is done deliberately. I think the author should clarify this in the intro. In the task, they are forced to make a decision (seek vs. not seek), but in real-life this might not be the case. People may avoid knowing whether another person has behaved unfairly without even realizing it. I think the question about people's belief the authors have added to the task is a clever strategy to solve this issue. However, they should enter this into their analyses as independent variable to understand whether participants will avoid information because they expect to receive something unfair (see below).

The distinction between deliberate ignorance and information avoidance is an important one that had so far been missing in our manuscript, and it is true that in real-life, many choices not to know can be unconscious. Following your feedback, we have specified this in the introduction (p.2, para. 2) by highlighting that deliberate ignorance is only a subset of information avoidance, where many cases are unconscious: “While information avoidance may be unconscious (Golman et al., 2017; Sharot & Sunstein, 2020), deliberate ignorance requires conscious choice. This can occur for strategic reasons [...]” Moreover, we are grateful for your suggestions for further addressing this concern by entering beliefs into the analyses. We outline how we incorporated your suggestions regarding the analyses of different psychological motives below (p.5, point 5).

3) Uncertainty, Ignorance, and Punishment

Now, in the planned analyses, if I have understood correctly, the authors are using uncertainty as a proxy for informational states (which in fact it is) and they want to see whether uncertainty influence punishments. That’s interesting but I do not think this will directly answer the research question. My understanding is that the authors want to know whether people avoid costly punishments by avoiding information. But how the authors can do so if they are not considering the subjects’ information-seeking choices? Off-course, with the uncertain condition they are modulating participants’ informational state but the only thing they can conclude with that analysis is that enhancing uncertainty reduces punishments, not that participants are avoiding unfairness information to avoid costly punishments. First, it is possible that participants avoid information just because they do not feel to reduce their uncertainty, and not that they do want to avoid unfairness (see below how these two quantities are actually correlated in the task). There is an intense literature on different motives underlying human information-seeking decisions the authors can refer to (e.g., Kobayashi et al. 2019, NHB, Sharot & Sunstein, 2020, NHB, Cogliati Dezza et al. 2022, Cognition etc..). Second, Q3 can answer whether ignoring can lead to reduced punishments. However, this doesn’t mean that people ignore info to avoid punishments. It just means that not having the information will obviously reduce punishments because people do not know whether the responder was fair or not.

Thank you for raising these important concerns. The model that we propose for punishment under uncertainty in the UG (see p.5, figure 1) is an outcome-based model and not a process-based model. In particular, we are interested in answering the questions whether uncertainty about inequality affects punishment (RQ1), whether inequality moderates the effect of uncertainty on punishment (RQ2), and whether ignorance predicts punishment under uncertainty (RQ3). Given these three questions, all the hypotheses that we test (see p.7, table 1) are outcome-based hypotheses and not process-based hypotheses. While responses to our questions on beliefs and psychological motives for seeking and not seeking information allow us to run exploratory analyses on possible processes, they are not central to our confirmatory analyses on outcomes. Furthermore, we understand your concern about a possible null effect for RQ2 on the basis of inter-individual differences on a population level, which is why we propose to collect and analyze additional data on the reasons for why participants choose to seek and choose not to seek information for the case of a null effect for RQ2 (for details, see below: p.5, point 5).

4) Alternative Analyses

I think to answer the research questions the authors need to test 1) whether people ignore information about unfairness – the predicted variable should be information-avoidance (yes, no) and inequality as the predictor (prediction: more avoidance with higher inequality) ; 2) whether this is done deliberately i.e., subjects report that they avoided information because they expected unfairness (for example, a mediation analysis on the relationship between information avoidance and unfairness might help with that) 3) the authors' Q3 - whether avoiding information leads to reduced punishments; 4) whether that relationship is mediated by people's belief. [If the authors decide to run this latter analysis their belief question should of course be asked before the information-seeking decisions. If the authors do not want to interfere with the other variables of the task, this can be run in an independent task Q3]

Thank you very much for your suggestions to further examine underlying processes on the basis of the belief data. As mentioned above (p.3, point 3), we are particularly interested in identifying outcomes and not processes. As such, our model has not been designed to examine underlying processes, and our statistical tests first and foremost focused on outcome-based hypotheses.

We appreciate your four suggestions for answering research questions with an additional focus on underlying processes. Suggestion #3 already corresponds to our statistical model #3 as a primarily outcome-based model. We think that the remaining three, more process-based tests will not benefit the design for the following three reasons. First, predicting ignorance by inequality will not necessarily answer the question whether people ignore information about unfairness since people can also ignore information about unfairness in a way that is independent from inequality. For example, 50% of participants may be ignorant in both treatment 2 and treatment 4. Here, inequality may not predict ignorance even though people still ignore information about unfairness. Note that this example does not contradict H2A since an interaction effect between uncertainty and inequality can still occur even when ignorance levels for high and low inequality are approximately even. Second, we think that the suggested mediation analysis on the relationship between ignorance and inequality will not benefit the design as we think that an analysis of the underlying main effect will not benefit the design (see reason described above). Third, we think that the suggested mediation analysis for the effect of ignorance on punishment will not enhance the design as we can expect zero variance in the beliefs of participants who choose to seek information. In particular, if participants choose to seek information and find out that the endowment is 100¢, they can be expected to believe that the endowment is 100¢. In contrast, only participants who choose not to seek information can believe that the endowment is not 100¢ but 60¢ or 20¢. Given that variance in beliefs can only be expected for ignorant but not for non-ignorant participants, we do not think that a mediation analysis for the effect of ignorance on punishment will yield helpful insights. Taken together, we are grateful for the additional suggestions on process-based questions, while our model and statistical tests focus first and foremost on outcome-based hypotheses.

5) Individual Differences

The other problem that I see is that in the current version of the task inequality and uncertainty (not the authors' uncertainty but the variance between the two offers) are the same thing. In other words, when participants are presented with treatment 4 they are both more uncertain about the possible offer and might incur in higher inequality compared to treatment 2. This means that the valance of the information (level of unfairness) and the uncertainty are anticorrelated. This can be problematic because participants may want to avoid information because it may reveal unfairness but they do also have the urge to reduce their uncertainty. Now, people who will be more prone to reduce their uncertainty, they might seek more info. People who will be more prone to avoid unfairness they might avoid that information. This might result in a null effect on a population level, without, however, being true. Which are the strategies the authors are planning to use to solve this issue?

The possibility of a null effect for RQ2 on the basis of inter-individual differences in information search is an important alternative explanation for H20. We have added this line of reasoning as an alternative explanation for a data pattern running contrary to our prediction to our manuscript (p.8, para. 3): "There are **two arguments** in favor of H20. [...] **Second, higher inequality comes, by definition, with a greater difference between fair and unfair offers. This might induce more information search for some, possibly counteracting less information search due to higher inequality for others – resulting in a null effect on the population level** due to inter-individual differences."

Should we fail to reject H20, we plan to assess this alternative explanation on empirical grounds by collecting both qualitative and, based on your important concern, additional quantitative data:

- First, we will collect qualitative responses to the questions "Why did you decide to find out [...]" and "Why did you decide not to find out [...]" by participants who decided to seek information and decided not to seek information, respectively.
- Second, we will ask participants how strongly they agree to either of the two statements (dependent on their choice behavior (not) to seek information): "I chose to find out as I wanted to know about possible inequality" or "I chose not to find out as I did not want to know about possible inequality" on seven-point Likert scales.

If we fail to reject H20, we will analyze the quantitative responses in a confirmatory way. Specifically, if the alternative explanation is correct, then we would expect that the motives for seeking information due to uncertainty aversion (here in accordance with your definition of uncertainty as the variance between offers) and not seeking information due to avoiding inequality cancel each other out, such that the agreement statements (ω) do not predict ignorance (v) after controlling for inequality (z). The regression model is the following:

$$(0) \quad v_i = \beta_0 + \beta_1 \omega_i + \beta_2 z_i + \varepsilon_i$$

For this model, we define the null hypothesis (H0) as ω not predicting v after controlling for z . Our alternative hypothesis (H1) is that ω predicts v after controlling for z . For the

significance level, we use $\alpha = 0.05$. A rejection of the null hypothesis will be interpreted as evidence against the alternative explanation that motives for seeking information and not seeking information cancel each other out, and in favor of the initial interpretation that responders behave in line with the Fehr and Schmidt model.

In case we fail to reject both H20 and H0, we plan to further assess your alternative explanation by examining qualitative responses to the questions “Why did you decide to find out [...]?” and “Why did you decide not to find out [...]?”. Coding these responses will allow us to assess whether there is any evidence for inter-individual differences that could explain a null effect on the population level. Concretely, we suggest that two independent raters code all responses to the two aforementioned questions for either the presence of a motive of uncertainty aversion (here, again, your definition of uncertainty as the variance between two offers) or a motive of avoiding unfairness, in a binary way. Both raters would be blind to the research question, and, in case of disagreement, an independent third party would decide whether a response can be coded as indicating the presence of a motive of uncertainty aversion or a motive of avoiding unfairness in cases where individuals decide to seek and decide not to seek information, respectively.

6) Improving Bonus Payments

Related to this point, I'm not sure I have all the details to evaluate this, but it seems to me that the information can be useful for participants to improve their bonus payment. Is it? Let's say that a participant is in treatment 4, if they seek information they can reveal that the proposer has received 100. They receive 50 and they accept the offer. The information has helped the participant to add 50 to their bonus payment. Now, if they didn't have that information, they may reject the offer and therefore lose 50. This is another confound in the task. People will seek information, regardless of uncertainty and unfairness, because that information will help improving their bonus payment. Again, how are the authors planning to deal with this issue?

Thank you for sharing this thought. In our design, it is not the case that information can be instrumental for participants to improve their bonus payment. That is, participants always receive the largest payment by simply accepting the offer, independent of revealing information regarding the fairness of the offer.

7) Always Seeking Information

Lastly, the authors confront seeking (free) information with the cost of punishing others. But this implies that they avoid that information because they are making the reasoning if I obtain that info, I have to inflict cost (to themselves and to the other). But again, this might not be the case. Let's say they are not making that reasoning, the authors may find they always seek information, regardless of the inequality, because there is no cost in seeking that information.

Thank you for raising this important concern. Our second pilot study addressed this concern by implementing instructions based on Dana and colleagues (2007) as described by Grossman (2014). With this design which retains the possibility to reveal information without incurring a cost, 64% of participants chose not to seek information. In contrast, when additional costs for seeking information were introduced, 97% of participants chose not to seek information. This suggests that neither a floor nor a ceiling effect in ignorance will be expected with the Grossman-inspired task instructions based on Dana and colleagues, and that an additional introduction of costs for seeking information can be expected to lead to a ceiling effect in ignorance.

Minor

8) “Sheds” vs. “Might Shed”

“Introducing uncertainty about disadvantageous inequality (e.g., whether 2\$ or 10\$ have been split), with a corresponding option to freely seek information, sheds light on the role of deliberate ignorance in the context of punishment.” I think this is a hypothesis. “sheds”  “might shed”

Thank you for highlighting this important nuance, which we have incorporated by adjusting “sheds” to “might shed” in the manuscript (see p.4, para. 2).

9) Definition of Ignorance

“Deliberate ignorance is defined as a responder’s conscious and irreversible choice not to seek information on the amount of money the proposer has received.” Why irreversible??

We had initially defined deliberate ignorance as a conscious and irreversible choice as the option to seek information only occurs in a unique moment in time in our paradigm. If participants choose not to seek information at this moment, they cannot go back. Hence, the choice not to seek information is irreversible. At the same time, we see value in aligning our definition with the definition of deliberate ignorance that is also used in broader contexts and have therefore removed “irreversible” from the definition in our manuscript (see p.4, para. 3).

10) Enabling Ignorance

Figure 1 is not clear how uncertainty can enable ignorance. By definition, uncertainty enables seeking information to reduce ignorance. In the context of this behavioral task, if there is uncertainty subjects may decide or not decide to resolve it... but it doesn’t mean that it enables ignorance.

Thank you for highlighting another important nuance. We have addressed this concern by replacing “uncertainty enables ignorance” by “uncertainty allows for ignorance” in our

manuscript (see p.5, figure 1) since ignorance can only occur under uncertainty but is not necessarily enabled by it.

11) Effect Size Computation

Table 1. How did the authors compute the effect size? It would be useful to have pilot data to estimate the effect size.

This is a very important concern for which we have also received and incorporated additional editorial guidance. In particular, we have redone all of our power analyses with effect size estimates resting on two different sources of cumulative evidence from the literature. First, based on 17 UG studies with varying levels of inequality (Camerer, 2011), we have estimated the expected probabilities of rejection for high and moderate inequality under certainty as 68% and 28%, respectively. Second, based on a recent meta-analytic review of 22 studies on deliberate ignorance (Vu et al., Forthcoming), we have estimated the expected probability of ignorance for high and moderate inequality as 44% and 40%, respectively. Together, these two sources of cumulative evidence combined with an expected upper limit of 10% for rejections under ignorance (grounded in our pilot data) allow us to estimate the expected probabilities of rejection for high and moderate inequality under uncertainty as 41% and 18%, respectively – providing us with effect size estimates for all three research questions. The R code for our power analyses has been made available at the OSF (see code availability statement; p.17, para. 5) to support reproducibility.

12) Figure 3 – Information Seeking

Figure 3 should also have the moment in which subjects decide whether to seek or not information as it is right now, it seems that subjects doesn't make an info-see choice

Thank you for another important point. The moment where subjects decide whether to seek information or not to seek information occurs in step 3. We have reformulated this step as a “choice to reveal or not to reveal the endowment” in our manuscript (see p.13, figure 3) to address your concern and to support greater clarity.

Reviewer #2 Feedback:

0) Study Summary

The authors register a study about uncertainty and unfairness in an economic-social decision-making task. They aim to examine whether uncertainty about (the amount of) inequality influences their decision to punish inequality, whether this process is itself influenced by the amount of inequality, and whether consciously staying ignorant of the amount of inequality reduces the propensity to punish inequality. For this, they developed an experimental between-subjects design which introduces uncertainty about the initial

endowment of the proposer in an ultimatum game and the possibility for the responder to receive information about it.

The research plan seems solid: the hypotheses are well grounded in theory and well explained; the experimental design fits the hypotheses; the power analyses are reasonable and reproducible. However, there are some specifications/details missing or ambiguous which I list below and would suggest the authors clarify before conducting the study.

Thank you very much for your positive assessment of our research plan.

1) “Over-power” Analyses

The large sample required to answer RQ2 will “over-power” the analyses for RQs 1 and 3. This is not a problem in itself, but it implies that even smallest deviations between sample means (RQ1) or from an effect size of 0 (RQ3) will easily become statistically significant. Thus, I would recommend adding an estimate of the necessary effect size to be meaningful in your opinion (for research, for every-day life – however you want to define “meaningful”). This is aimed at thwarting interpreting statistically significant effects, which are so small that they are practically meaningless.

Thank you very much for raising this important concern on “over-powering” our analyses for RQ1 and RQ3. We have addressed this concern by adding the following section to our sampling plan (p.14, para. 4): “Given our power analyses, we see the possibility of ‘**over-powering**’ our analyses for RQ1 and RQ3. To avoid interpreting very small differences that are statistically significant but neither theoretically nor practically relevant, we commit ourselves to only interpreting effect sizes of 10% or more of our derived effect sizes (see Table 1) as being meaningful. That is, we will **interpret effect sizes of $d_1 < 0.04$ and $d_3 < 0.11$ for RQ1 and RQ3, respectively, as too small to be of theoretical or practical relevance.**” Please note that these effect sizes are based on new estimations grounded in cumulative evidence from the literature. The R code for our power analyses is available at the OSF (see code availability statement; p.17, para. 5) to support reproducibility.

2) Exclusion Criteria

The next issue regards the exclusion criteria. For the second exclusion criterion mentioned in the Sampling Plan section, please list the specific criteria explicitly that you will use so that readers do not have to read another article (Meade & Craig, 2012) to know the full set of exclusion criteria. Furthermore, you state to ask participants three comprehension questions after the experiment and exclude all those “who fail to answer all three comprehension questions correctly within two attempts”. Does that mean they have to fail all of the three questions to be excluded, that is, they are not excluded if they correctly answer one or two out of three, or are they excluded if they fail any one of the questions (within two attempts)? Please be as precise as possible in the description of any participant exclusion criteria. Furthermore, I would recommend being strict in exclusions due to possible misunderstanding of the instructions. The used task seems to be a one-shot experiment so that each participant answers the crucial trial just once. I understand the rationale for this

due to possible feedback effects. (This might be specified explicitly in the Methods section, though.) However, this design might make findings more susceptible to spurious effects of the wording of instructions and task understanding than designs with numerous trials per participant.

Thank you very much for your three points of improvement on our exclusion criteria, which we have implemented as follows. First, we have specified the second exclusion criterion by adding an explicit and unambiguous condition to our manuscript (marked in **bold**; p.15, para. 2): “Second, we will exclude participants who self-report careless participation. In particular, **we will not include choices by participants who answer ‘no’ to the question: ‘Honestly, should we use your data in the analysis of our study?’**”. Second, we have clarified that of the two options that you present for the comprehension check, we mean the latter: “For their data to be included in the analysis, participants have to answer all three questions correctly within two attempts. That is, **all participants who still give at least one wrong answer in their second attempt will be excluded** from our analysis”. This is also in line with your recommendation to be strict with exclusion. Specifically, based on these criteria, our exclusion rates for pilots 1 and 2 were 12% and 16%, respectively. Finally, we have specified the one-shot nature of our study design and the corresponding rationale for this study design in our methods section (p.12, para. 1), in line with your recommendation: “We start our investigation whether deliberate ignorance influences punishment behavior by relying on a **one-shot punishment setting**. We do so mainly **to study causal links in the absence of feedback**.”

3) Analysis Plan

Your chosen alpha level (0.05) is implied in the sampling plan in Table 1, but it would be good to specify it explicitly in the Analysis Plan section of the Methods. Furthermore, please report the specifics of any preceding or accompanying analysis you might have planned in more detail (e.g., evaluating the overall model fit/explained variance of the regression models before interpreting regression coefficients; restrictions put on the regression coefficients/procedure how to handle regression coefficients smaller than 0 or larger than 1 (which might happen with linear probability models); a priori planned or criteria for post hoc inclusion of control variables in the analyses; testing the assumed heterogeneity of variance for analysis of RQ2).

Thank you for your ideas on how to further strengthen our analysis plan. Following your recommendation, we have specified our alpha level explicitly in our analysis plan (p.15, para. 3): “For the significance level, **we use $\alpha = 0.05$ for RQ1, RQ2, and RQ3** to test our preregistered hypotheses.” Furthermore, we have implemented your suggestion to specify any preceding or accompanying analysis by inserting the following section: “The **requirements for the three regression models** will be verified in preceding analyses. In particular, we will assess whether **predictions of less than 0.05 or more than 0.95** are made by our linear probability models. If such predictions occur, we will report additional logit models next to our respective linear probability models to support the robustness of our estimates. Before we interpret β_{11} , β_{23} , and β_{31} , we will **assess the overall model fit of our three regression models** in terms of explained variance on the basis of $\alpha = 0.05$. We do not

plan any further **post hoc inclusions of control variables** on the basis of our preregistered hypotheses” (p.16, para 2). Finally, we have added an analytic solution for the derivation of the expected variances for the rejection rates across the four treatments on the basis of the underlying Bernoulli distributions to increase the robustness of our variance estimates for the power analysis for RQ2 (p.14, para. 2).

4) Sample Description

I would encourage you to add a few questions for the sake of sample description. As research often relies on WEIRD (Western, educated, industrialized, rich, democratic) samples but online studies offer the chance to reach a more diverse sample than studies on-site at universities, it would be interesting for the interpretation and possible later aggregation of this data if you would report some sample characteristics (e.g., age, sex/gender, education level, ethnicity). [With this, I do not aim to encourage you to add these variables in the inference statistical tests but merely use them descriptively for sample characterization.]

Thank you very much for this important recommendation on the reporting of demographics. We have included questions on all your suggested characteristics (i.e., age, sex/gender, education level, and ethnicity) in our design for the pilot studies and will continue to include these questions in our main study. The sample descriptions for both of our pilot studies are reported in detail in the Supplementary Information (p.2, para. 2; p.4, para. 4). Sample characteristics for our main study will be reported in the stage 2 manuscript.

Reviewer #3 Feedback:

0) Study Summary

This registered study entitled “Avoiding Punishment by Ignoring Unfairness: Responses to Uncertain Inequality in the Ultimatum Game” (COMMSPSYCHOL-23-0037-T) plans to investigate the effect of deliberate ignorance on the punishment (i.e., reject) in the Ultimatum Game (UG). The UG is an established task to investigate the unfairness and inequality in distribution situations. Deliberate ignorance is a novel factor that influences the response to an offer (accept or reject) in the UG. Hence, this study will be of interest to researchers in related areas. The background and motivation for this study are clear to me, and the sampling plan (i.e., the number of participants based on the assumed effect size) seems valid. However, methods of data analysis are not well described. Therefore, I would like the authors to respond to my concerns below.

Thank you very much for your interest in our study and for your positive assessment of the background, motivation, and sampling plan for our study. We hope that our revisions to the manuscript and our point-by-point responses will fully address your concerns on our description of the data analysis plan.

1) Model Comparison

How many models can be considered in total, and how do they compare (e.g., Akaike information criteria)?

Three models will be considered in total for which we have added the following specifications in our revised manuscript (p.15-16):

$$(1) \quad y_i = \beta_{10} + \beta_{11} x_i + \varepsilon_i$$

$$(2) \quad y_i = \beta_{20} + \beta_{21} x_i + \beta_{22} z_i + \beta_{23} x_i z_i + \varepsilon_i$$

$$(3) \quad y_i = \beta_{30} + \beta_{31} v_i + \varepsilon_i$$

In these three models, y refers to punishment, x to uncertainty, z to inequality, and v to ignorance. The error term is specified as ε_i for individuals $i = 1, \dots, n$. For all three models, we will first assess the overall model fit in terms of explained variance based on $\alpha = 0.05$ before interpreting β_{11} , β_{23} , and β_{31} to answer our three research questions.

2) Analysis Method

Related to the first concern, what methods do you use in your data analysis? Path analysis? Mediation analysis?

Do the authors use a logit model to fit the punishment behavior?

For all our tests, we rely on linear probability models as the interaction effect herein corresponds to the marginal effect of the interaction term, unlike interaction effects in logit models. Our three statistical models are specified above (see equations 1-3), and we do not plan to conduct any mediation or path analysis.

3) Figure 1

Can the authors indicate which arrow corresponds to which hypothesis? This is a minor point.

Thank you very much highlighting that we had not yet sufficiently specified the correspondence between our hypotheses and the arrows in Figure 1 (p.5). We have made the links to the hypotheses explicit in the figure's annotation to clarify the correspondence without convoluting the diagram. In particular, we have addressed this important concern as follows in the description of Figure 1: "We expect a lower probability of punishment under uncertainty (**H1A; hypothesis for RQ1**). Inequality is expected to moderate this negative relationship between uncertainty and punishment (**H2A; hypothesis for RQ2**). Since uncertainty allows for ignorance, we expect that ignorance reduces the probability of punishment (**H3A; hypothesis for RQ3**)".

28th Sep 23

Dear Mr Offer,

Thank you once again for your manuscript, entitled "Avoiding Punishment by Ignoring Unfairness: Responses to Uncertain Inequality in the Ultimatum Game," and for your patience during the peer review process.

Your revised manuscript has now been evaluated by the 3 original reviewers and they are all in principle satisfied with your revisions. Before we make a decision on in principle acceptance and Stage 2 submission, we would like you to address the minor concerns raised by reviewer 2 in a final revision.

Editorially, we would also like you to clarify one aspect of the uncertainty manipulation. Initially there is an uncertainty manipulation (e.g., providing or not providing the information about the proposer size of the pie), but as soon as the participants in the uncertain condition choose to know about know about the proposer size of the pie, the uncertainty disappears, so all dependent variables that are measured after that event (choice to reject, choice to punish) would need to take that into account. From our understanding of the experimental design, ignorance and uncertainty are not independent from each other. In the sense that the manipulation of uncertainty is valid on all the dependent variable, only if participants choose to remain ignorant. The pilot data does provide valuable information on this point, showing that there is at least 25% of the sample that choose to remain ignorant. If this is the case, this aspect will need to be integrated in the analytical plan.

In sum, we invite you to undertake a final revision your Stage 1 Registered Report addressing the last remaining concerns. Please highlight all changes in the manuscript text file.

* Include a "Response to reviewers" document detailing, point-by-point, how you addressed the referee comment.

* In your cover letter, please include the following information:

--An anticipated timeline for completing the study if your Stage 1 submission is accepted in principle.

--A statement confirming that you agree to share your raw data, any digital study materials, computer code, and laboratory log for all eventually published results.

--A statement confirming that, following Stage 1 in principle acceptance, you agree to register your approved protocol on the Open Science Framework (<https://osf.io/>) or other recognised repository, either publicly or under private embargo, until submission of the Stage 2 manuscript.

--A statement confirming that if you later withdraw your paper, you agree to the Journal publishing a short summary of the pre-registered study under a section Withdrawn Registrations.

Please use the link below to submit your final revised manuscript and related files:

[link redacted]

We hope to receive your final revised manuscript within two weeks. If you cannot send it within this time, please let us know. We will be happy to consider your revision so long as the report still represents a significant contribution to the literature at that stage.

* **TRANSPARENT PEER REVIEW:** Communications Psychology uses a transparent peer review system. This means that we publish the editorial decision letters including Reviewers' comments to the authors and the author rebuttal letters online as a supplementary peer review file. We publish these records for all accepted manuscripts. However, on author request, confidential information and data can be removed from the published reviewer reports and rebuttal letters prior to publication. If your manuscript has been previously reviewed at another journal, those Reviewers' comments would not form part of the published peer review file.

Communications Psychology is committed to improving transparency in authorship. As part of our efforts in this direction, we are now requesting that all authors identified as 'corresponding author' on published papers create and link their Open Researcher and Contributor Identifier (ORCID) with their account on the Manuscript Tracking System (MTS), prior to acceptance. ORCID helps the scientific community achieve unambiguous attribution of all scholarly contributions. You can create and link your ORCID from the home page of the MTS by clicking on 'Modify my Springer Nature account'. For more information please visit www.springernature.com/orcid.

We look forward to seeing the final revised manuscript and thank you for the opportunity to review your work.

Sincerely,

Eva R. Pool, PhD
Editorial Board Member
Communications Psychology
orcid.org/0000-0001-5929-1007

REVIEWERS' COMMENTS:

Reviewer #1:

Remarks to the Author:

I appreciated the effort made by the authors in revising the manuscript and addressing my concerns. The authors have addressed all my concerns.

Reviewer #2:

Remarks to the Author:

The authors have addressed my previous concerns very well. The added two pilot studies provide important pieces of information about the task/study design; the added information in the Methods section in response to my and the other reviewer's comments increase the clarity and comprehensibility of the manuscript greatly. I have only two minor points I would like the authors to add to the manuscript.

1. What exactly does the "hidden information" treatment in Grossman's (2014) operationalization of Dana's (2007) approach mean? This is about what you write in the Supplement pages 3-4 about what kind of manipulations earlier studies used but you did not in Pilot Study 1. I think it would be beneficial for readers to explicitly state in the Methods section of the manuscript which of those manipulations you used in Pilot Study 2 and will use in the actual study of the Registered Report so that any reader knows without a doubt how exactly your study procedure worked.
2. If you check the overall model fit of the regression models, as you state now on page 19 of the manuscript, and this test does not show a statistically significant amount variance explained, will you still interpret the regression coefficients or discard the model altogether? Please add this information to the analysis plan.

References

- Dana, J., Weber, R. A. & Kuang, J. X. Exploiting moral wiggle room: Experiments demonstrating an illusory preference for fairness. *Economic Theory* 33, 67–80 (2007).
- Grossman, Z. Strategic ignorance and the robustness of social preferences. *Management Science* 60, 2659–2665 (2014).

Reviewer #3:

Remarks to the Author:

The authors have addressed all my concerns.

COMMUNICATIONS PSYCHOLOGY – STAGE 1 RR

POINT-BY-POINT RESPONSES TO REVIEWS

Avoiding Punishment by Ignoring Unfairness:

Responses to Uncertain Inequality in the Ultimatum Game

Konstantin Offer, Dorothee Mischkowski,

Zoe Rahwan, Christoph Engel

09 October 2023

Reviewer #2 Feedback:

1) Operationalization “Hidden information” treatment

What exactly does the “hidden information” treatment in Grossman’s (2014) operationalization of Dana’s (2007) approach mean? This is about what you write in the Supplement pages 3-4 about what kind of manipulations earlier studies used but you did not in Pilot Study 1. I think it would be beneficial for readers to explicitly state in the Methods section of the manuscript which of those manipulations you used in Pilot Study 2 and will use in the actual study of the Registered Report so that any reader knows without a doubt how exactly your study procedure worked.

Thank you very much for bringing this important concern to our attention. We have addressed this important concern by adding in-depth descriptions of all four manipulations in our pilot study 2 – including condition two, the “hidden information” treatment in Grossman's (2014) operationalization of Dana et al. (2007), which will be used in our main study. More precisely, we have added the following specifications (additions highlighted in **bold**) to our methods section in the manuscript (p.17, para. 2-3):

The second pilot study examined information search for varying instructions and costs for seeking information (see also SI2, Figure SI-2). **The pilot study consisted of four conditions, all of which had moderate inequality and uncertainty. The first condition was designed as a control condition employing the same instructions as in the first pilot study. In particular, participants in this condition were not told how the proposer’s endowment had been determined, whether the other person would be informed about their information seeking or not, and whether the interaction would be anonymous or not. Information on the proposer’s endowment could be sought by clicking a button labelled “reveal other person’s money” – making “no reveal” the default choice as in previous studies (Bartling et al., 2014; Dana et al., 2007; Feiler, 2014; Grossman & Van der Weele, 2017). The second condition employed instructions as described by Grossman (2014). In particular, participants in this condition were told that the endowment had been randomly determined by a computer, that the other person would not be informed about their information seeking, and that the interaction would be anonymous. Information on the proposer’s endowment could be sought by selecting one of two buttons labelled “Proceed” and “Reveal version”, with the “Proceed” button preselected – making “no reveal” the default choice, as in previous studies and condition one. Conditions three and four were identical to condition two with the only difference that they introduced additional costs of 10¢ and 20¢ for seeking information, respectively.**

The mean probabilities of ignorance for conditions one, two, three, and four were 25%, 64%, 95%, and 100%, respectively. These findings suggest that costs for seeking information can be expected to lead to a floor effect in information search, and that **the instructions employed in condition two (where participants are informed about how the proposer’s endowments had been determined, whether the other person**

would be informed about their information seeking or not, and whether the interaction would be anonymous or not) can be expected to neither lead to a floor nor a ceiling effect in information search. **Based on these findings, instructions from condition two will be used in the main study to employ a study design for which neither floor nor ceiling effects can be expected.**

Furthermore, we have uploaded all of our instructions for both pilot study 1 and pilot study 2 to OSF (see here) to make them publicly available to support transparency and reproducibility.

References

- Bartling, B., Engl, F., & Weber, R. A. (2014). Does willful ignorance deflect punishment? – An experimental study. *European Economic Review*, 70, 512–524.
<https://doi.org/10.1016/j.euroecorev.2014.06.016>
- Dana, J., Weber, R. A., & Kuang, J. X. (2007). Exploiting moral wiggle room: Experiments demonstrating an illusory preference for fairness. *Economic Theory*, 33(1), 67–80.
<https://doi.org/10.1007/s00199-006-0153-z>
- Feiler, L. (2014). Testing models of information avoidance with binary choice dictator games. *Journal of Economic Psychology*, 45, 253–267.
<https://doi.org/10.1016/j.joep.2014.10.003>
- Grossman, Z. (2014). Strategic ignorance and the robustness of social preferences. *Management Science*, 60(11), 2659–2665. <https://doi.org/10.1287/mnsc.2014.1989>
- Grossman, Z., & Van der Weele, J. J. (2017). Self-image and willful ignorance in social decisions. *Journal of the European Economic Association*, 15(1), 173–217.
<https://doi.org/10.1093/jeea/jvw001>

2) Overall fit of the regression models

If you check the overall model fit of the regression models, as you state now on page 19 of the manuscript, and this test does not show a statistically significant amount variance explained, will you still interpret the regression coefficients or discard the model altogether? Please add this information to the analysis plan.

Thank you very much for highlighting a lack of specification with regard to the interpretation of our regression coefficients in the case of an insignificant regression model. We have addressed this concern by committing ourselves to not interpreting regression coefficients should an entire model turn out insignificant. More specifically, we have added the following specification to our methods section in the manuscript (p.16, para. 2): **“If the overall model fit of a regression model is not statistically significant, we will not interpret any regression coefficients and discard the model altogether.”**

13th Oct 23

Dear Konstantin,

Thank you once again for submitting your revised Stage 1 Registered Report, entitled "Avoiding Punishment by Ignoring Unfairness: Responses to Uncertain Inequality in the Ultimatum Game." Everything is in order and I am delighted to say that we can offer acceptance in principle. You may progress to Stage 2 and complete the study as approved.

As you know, a condition of in-principle-acceptance is that the authors agree to deposit their Stage 1 accepted protocol in a repository, either publicly or under embargo until Stage 2 acceptance and publication. We are very keen to showcase our in-principle accepted protocols, so that our readers, reviewers, and potential authors can gain insight into the requirements of the format as well as an idea of the types of projects that are suitable for publication in Communications Psychology. We have set up a space on figshare to host all of our in-principle accepted protocols, which can either be made public or kept under embargo until Stage 2 acceptance (depending on author preference). This gives you the opportunity to have your work publicly associated with Communications Psychology, and of course we will be very pleased to showcase your report if you agree to share it publicly.

Depositing the work on our figshare space does not preclude deposition of your Stage 1 protocol on other depositories – your protocol can also be posted on OSF, Dataverse, Dryad or any other public repository of your choice. You also do not need to do anything – if you agree with posting your protocol on our figshare space, we will upload your protocol on your behalf and either set it public or place it under embargo, depending on your choice. Your protocol will be licensed under a CC BY license (Creative Commons Attribution 4.0 International License). The CC BY license allows for maximum dissemination and re-use of open access materials and is preferred by many research funding bodies. Under this license users are free to share (copy, distribute and transmit) and remix (adapt) the contribution including for commercial purposes, providing they attribute the contribution in the manner specified by the author or licensor (read full legal code: <http://creativecommons.org/licenses/by/4.0/legalcode>) Please note that any use of <https://springernature.figshare.com> will be subject to the Figshare terms of use. Figshare has the right to enforce these terms and conditions where applicable. Use of third party services and sites will be subject to the relevant terms of use and will apply if we act on your behalf in this regard. Do let me know if you would like to take up this option or if you have any questions regarding the protocol deposition requirement.

IMPORTANT:

In cases where the registered experimental design is altered after AIP due to unforeseen circumstances (e.g. change of equipment or unanticipated technical error), the authors should consult the editors immediately for advice, prior to the completion of data collection.

Following completion of your study, we invite you to resubmit your paper for peer review as a Stage 2 Registered Report. Please note that your manuscript can still be rejected for publication at Stage 2 if the Editors consider any of the following to hold:

- The results were unable to test the authors' proposed hypotheses by failing to meet the approved outcome-neutral criteria
- The authors altered the Introduction, rationale, or hypotheses, as approved in the Stage 1

submission

- The authors failed to adhere closely to the registered experimental procedures without previously seeking editorial approval
- Any post hoc (unregistered) analyses were either unjustified, insufficiently caveated, or overly dominant in shaping the authors' conclusions
- The authors' conclusions were not justified given the data obtained

We encourage you to read the complete guidelines for authors concerning Stage 2 submissions at <https://www.nature.com/commspsychol/submit/registered-reports> and <https://www.nature.com/documents/commspsychol-style-formatting-checklist-article-rr.pdf>.

Please especially note the requirements for protocol deposition, data sharing, and that withdrawing your manuscript will result in publication of a Retracted Registration.

When you are ready, please use the following link to access your home page and submit your Stage 2 Registered Report:

[link redacted]

*This url links to your confidential homepage and associated information about manuscripts you may have submitted or be reviewing for us. If you wish to forward this e-mail to co-authors, please delete this link to your homepage first.

* TRANSPARENT PEER REVIEW: Communications Psychology uses a transparent peer review system. This means that we publish the editorial decision letters including Reviewers' comments to the authors and the author rebuttal letters online as a supplementary peer review file. This means that the records will be published together with your Stage 2 report. On author request, confidential information and data can be removed from the published reviewer reports and rebuttal letters prior to publication. If your manuscript has been previously reviewed at another journal, those Reviewers' comments would not form part of the published peer review file.

We expect your Stage 2 Registered Report to be submitted by the date specified in your latest cover letter. If unforeseen circumstances prevent submission by that date, please contact us as soon as possible to discuss any changes to the submission time-frame.

Thank you again for offering us this work and we look forward to receiving your Stage 2 Registered Report.

Yours sincerely,
Marika, on behalf of

Eva R. Pool, PhD
Editorial Board Member
Communications Psychology
orcid.org/0000-0001-5929-1007

7th Mar 24

Dear Konstantin,

Thank you once again for submitting your Stage 2 Registered Report, entitled "Deliberately Ignoring Unfairness: Responses to Uncertain Inequality in the Ultimatum Game," and for your patience during the re-review process.

Your manuscript has now been evaluated by Reviewers #1, #2, and #3 from the previous rounds of review, whose comments are included at the end of this letter. In the light of our reviewers' advice, we are pleased to inform you that we will be able to accept your Stage 2 manuscript, pending revisions to address Reviewers' comments and editorial requests.

In detail, we ask that you revise the Discussion to reduce the speculation of potential causes of the null effects for hypotheses #1 and #2. We generally discourage speculation on matters that cannot be resolved and in this case, as the referee highlights, it is sufficient to re-state the a priori mentioned potential interpretation, but highlight that the matter remains unresolved.

The reviewers also include some guidance on presentation, which we encourage you to adopt for the Results and SI (Introduction and Methods may not change).

One of the main reasons for delays in eventual acceptance is failure to fully comply with editorial policies and formatting requirements. To assist you with finalizing your manuscript for publication, I attach our Editorial Requests Table which lists all of our editorial policies and formatting requirements.

Please attend to *every item* in the Table and upload a copy of the completed checklist with your submission. I have highlighted in the checklist items that require your attention.

OPEN ACCESS:

Communications Psychology is a fully open access journal. Articles are made freely accessible on publication under a CC BY license (Creative Commons Attribution 4.0 International License). This license allows maximum dissemination and re-use of open access materials and is preferred by many research funding bodies.

For further information about article processing charges, open access funding, and advice and support from Nature Research, please visit <https://www.nature.com/commpsychol/article-processing-charges>

At acceptance, you will be provided with instructions for completing this CC BY license on behalf of all authors. This grants us the necessary permissions to publish your paper. Additionally, you will be asked to declare that all required third party permissions have been obtained, and to provide billing information in order to pay the article-processing charge (APC).

* TRANSPARENT PEER REVIEW: Communications Psychology uses a transparent peer review system. On author request, confidential information and data can be removed from the published reviewer reports and rebuttal letters prior to publication. If you are concerned about the release of confidential data, please let us know specifically what information you would like to have removed. Please note that we cannot incorporate redactions for any other reasons.

We hope to hear from you within three weeks; please let us know if the revision process is likely to take longer.

Please use the following link for uploading the materials:

[link redacted]

Best wishes,
Marike

Marike Schiffer, PhD
Chief Editor
Communications Psychology

Reviewer #1:

Remarks to the Author:

I enjoy reading this manuscript again and interesting interpretation of the results. I have some minor comments to improve the readability and clarity of the result section as well as the discussion.

A table with the demographic information, instead of having them in the text would reduce the overload of written information in the text.

Line 495 "(II) ensure that we faced neither a floor nor a ceiling effect in the proportion of individuals 495 who choose not to seek information (i.e., $v = 1$)" I don't see these results in the paragraph below.

Line 501: Perhaps I'm wrong but shouldn't this sentence "Figure 4 displays the effect of uncertainty on punishment by inequality" go somewhere else? Here the authors are just discussing the inequality results.

Line 502: "linear probability model" Please specify again which linear model.

Overall figures could be slightly improved by removing the background grid and increasing the font size as well as adding a significance level and explaining it in the text (e.g., ** [n the figure] ** $p < 0.01$ [in the text]).

Figure 4. Shouldn't the X axis read as "informational state"? And then certainty and uncertainty as sub-categories? (I know the authors used informational state for something else in Figure 5, but it is very confusing to have "uncertainty" divided into "certain inequality" and "uncertain inequality". Y

axis may better read like this “Punishment - costly rejection of an offer”, to immediately understand what punishment refers to.

Line 507: maybe I would add a figure for the results reported here. Also, how did the author code “ignorance” (e.g., 0=certainty; 1=uncertainty)?

Generally, I feel that repeating some of the info provided in the methods can help to quickly grasp the results.

Line 537: Although they are reported in the supplementary material, it would be nice if the authors summarize the results of this “To analyze all cases and close this gap, we ran two additional analyses (see S15), 537 using a Bonferroni-adjusted $\alpha_{adj} = 0.05/3$ to account for multiple comparisons” in one line in the main text.

Line 541: I don’t think it is very clear what the authors did here “Regressing punishment on a dummy variable for the first and second 541 informational states, we found a significant difference in punishment ($\beta_{51} = 0.129$, SE = 0.022, $t(1037) = 5.796$, $p < 0.001$).”

Lines 540-548: the authors should also include the mean probability of punishment for the information-seeking responders.

Figure 5. Similar comments as those provided for Figure 4

Line 619: “but [we] could not reject”

Line 618: Shouldn’t be the other way around “We found a significantly higher probability of punishment by ignorant than by non-ignorant responders”? Punishment is defined as rejecting an unfair offer, isn’t it?

Do the authors’ results really support such an interpretation “We interpret these results as evidence for sorting behavior in that individuals select their informational states based on their social preferences” ? Can the authors better explain what they mean with “social preferences” here? Again, I don’t think the data support this claim “Those willing to punish seek information about unfairness, while those unwilling to punish ignore it to reduce cognitive dissonance and lower emotional costs.” It seems to me that participants first decided whether to know or not and then they decided to whether accept or reject the offer, didn’t they? If the above is correct, then the authors cannot have access to participants’ preferences for willingness to punish before they decide to seek information. Therefore, their data doesn’t really support their conclusion.

Reviewer #2:

Remarks to the Author:

Overall, the authors have adhered to their registered study design and analysis plan. As a pre-defined outcome-neutral criterion, floor and ceiling effects in participants’ behavior were checked.

Introduction, rationale, and hypotheses are the same as in the registration. They have added some exploratory analyses to explain some of the failed rejections of stated null hypotheses, which are clearly marked as such. The exploratory analyses take up more room in presenting Results and their

Discussion than the registered analyses. Moreover, the a priori stated interpretation of null effects for RQs 1 and 2 (“Reinforcement of a fairness norm based on altruistic punishment” and “Punishment in line with classic economics of information and economic theories of fairness”) receive comparatively little attention in the Discussion. However, the exploratory analyses were mostly done to illuminate possible mechanisms behind the hypothesized but not found effects, so this imbalance should probably not be interpreted too negatively. Yet I think, the paper might benefit from taking the initially stated interpretations more into account while discussing the findings. In sum, this is a well written paper providing interesting insights and a good discussion about a possible mechanism driving behavior under uncertainty and inequality possibly dependent on (not measured) personality traits. Besides the issue of interpreting the null effects of RQs 1 and 2, I have only two minor comments to the authors:

1. Please state the overall model fit analyses for the first two models addressing RQs 1 and 2, subsumed under the heading “Uncertainty and Punishment”. At the moment, these statistics are only given for RQ3.
2. In the section “Ignorance and Punishment” in the Results, you state a confidence interval for the proportion of participants who did not seek information. This information confused me. Why is there a standard error to calculate a CI with for a proportion you know directly from participants behavior?

Reviewer #3:

Remarks to the Author:

This registered study entitled “Avoiding Punishment by Ignoring Unfairness: Responses to Uncertain Inequality in the Ultimatum Game” (COMMSPSYCHOL-23-0037C) investigated the effect of deliberate ignorance of uncertainty on the punishment (i.e., reject) in the Ultimatum Game (UG). Their results showed that the influence of uncertainty on the rejection rate was not significantly different from that of inequality. No interaction between uncertainty and inequality was observed. Instead, the authors found that participants who sought the information of distribution rejected the offer more than those who ignored the information. These results suggest that participants who had the will to punish sought the information, while those who did not have the will to punish avoided revealing the information.

Enough participants were recruited based on the results of pilot studies. The analyses, although simple, are sufficient to test the hypotheses. The results differ in some respects from what the authors expected, but the authors report honestly. The authors’ interpretation of the results seems sound. Therefore, I have no concerns to be addressed.

COMMUNICATIONS PSYCHOLOGY – STAGE 2 RR

POINT-BY-POINT RESPONSES TO REVIEWS

**Deliberately Ignoring Unfairness: Responses to
Uncertain Inequality in the Ultimatum Game**

Konstantin Offer, Dorothee Mischkowski,

Zoe Rahwan, Christoph Engel

20 March 2024

Reviewer #1 Feedback:

0) Remarks to the Author

I enjoy reading this manuscript again and interesting interpretation of the results. I have some minor comments to improve the readability and clarity of the result section as well as the discussion.

Thank you very much for your positive evaluation of our Stage 2 manuscript and for your comments for further improving our results and discussion sections. We document below how we have incorporated your comments.

1) Demographic Information

A table with the demographic information, instead of having them in the text would reduce the overload of written information in the text.

We have removed the text and added a table. Thank you!

2) Second Manipulation Check

Line 495 "(II) ensure that we faced neither a floor nor a ceiling effect in the proportion of individuals 495 who choose not to seek information (i.e., $v = 1$)" I don't see these results in the paragraph below.

The results are reported in detail in lines 502 to 505: "Specifically, the ignorance rates under moderate and high inequality were 56.7% (95% CI [0.513, 0.619]) and 48.4% (95% CI [0.431, 0.538]), respectively. Hence, we [...] neither faced a floor nor a ceiling effect in the proportion of individuals choosing not to know."

3) Figure 4 Callout

Line 501: Perhaps I'm wrong but shouldn't this sentence "Figure 4 displays the effect of uncertainty on punishment by inequality" go somewhere else? Here the authors are just discussing the inequality results.

We have removed the sentence from its initial position and added the following sentence to the end of the "Uncertainty and Punishment" section (now line 543): "Figure 4 displays the effects of uncertainty and inequality on punishment." Thank you.

4) LPM Specification

Line 502: "linear probability model" Please specify again which linear model.

We have specified “linear probability model” as “linear probability model (1)”. For consistency, we have also added specifications for “linear probability model (2)” and “linear probability model (3)”. Thank you!

5) Figure Improvements

Overall figures could be slightly improved by removing the background grid and increasing the font size as well as adding a significance level and explaining it in the text (e.g., ** [n the figure] ** $p < 0.01$ [in the text]).

We have removed the background grid and increased the font size for Fig. 4, Fig. 5, Fig. SI-1, Fig. SI-2, and Fig. SI-3. Further, we have added significance levels for Fig. 5 and Fig. SI-1. We have explained these consistently in the figure texts, enhancing readability. Thank you.

6) Figure 4 Labels

Figure 4. Shouldn't the X axis read as “informational state”? And then certainty and uncertainty as sub-categories? (I know the authors used informational state for something else in Figure 5, but it is very confusing to have “uncertainty” divided into “certain inequality” and “uncertain inequality”. Y axis may better read like this “Punishment - costly rejection of an offer”, to immediately understand what punishment refers to.

We have changed the X-axis label to “Informational State” with “Certainty” and “Uncertainty” as sub-categories to ensure consistency across Fig. 4 and Fig. 5. We have further changed the Y-Axis label to “Punishment (costly rejection)” in Fig. 4, Fig. 5, SI-1, and SI-3. Thank you!

7) Added Figure / Ignorance Coding

Line 507: maybe I would add a figure for the results reported here. Also, how did the author code “ignorance” (e.g., 0=certainty; 1=uncertainty)?

We have coded ignorance as the choice to seek ($v=0$) or not to seek ($v=1$) information. To recap, ignorance can only occur under uncertainty ($x=1$; see lines 263–268). We could not add a figure in reference to the line 507 (of our initial stage 2 manuscript) since this line referred to the caption for Figure 4 (which was displayed in Line 506). Together, Figure 4 and Figure 5 display our primary findings for RQ1, RQ2, and RQ3.

8) Repeated Methods Info

Generally, I feel that repeating some of the info provided in the methods can help to quickly

grasp the results.

We have now repeated the following three pieces of information from our methods section in our “Uncertainty and Punishment” and “Ignorance and Punishment” sections:

1. “To recap, we tested our hypotheses for RQ1 by predicting the probability of punishment (y) by uncertainty (x).”
2. “To answer RQ2, we regressed punishment (y) on uncertainty (x), inequality (z) and the interaction term.”
3. “To examine whether ignorance predicts punishment under uncertainty (RQ3), we predicted the probability of punishment (y) by ignorance (v) for all participants within uncertainty treatments ($x = 1$).”

Repeating these three pieces of information has contributed to greater clarity and readability. Thank you.

9) Summary Additional Analyses

Line 537: Although they are reported in the supplementary material, it would be nice if the authors summarize the results of this “To analyse all cases and close this gap, we ran two additional analyses (see SI5), 537 using a Bonferroni-adjusted $\alpha_{adj}=0.05/3$ to account for multiple comparisons” in one line in the main text.

We have clarified the reference to SI5: “(for regression models, see SI5).” The results of the first additional analysis are summarized in lines 603–607. The results of the second additional analysis are reported in lines 607–627. Thank you for highlighting unclarity in the reference.

10) Specification Additional Analysis

Line 541: I don’t think it is very clear what the authors did here “Regressing punishment on a dummy variable for the first and second 541 informational states, we found a significant difference in punishment ($\beta_{51} = 0.129$, SE = 0.022, $t(1037) = 5.796$, $p < 0.001$).”

We have clarified our first additional analysis: “Our first additional analysis compares the mean probability of punishment for deliberately ignorant ($s_{d1} = 0$) and directly informed ($s_{d1} = 1$) responders. Regressing punishment on s_{d1} , we found [...]”. We have further made the reference to the regression model more explicit (see response to point 9 above). Thank you for bringing unclarity in the initial statement to our attention!

11) Punishment info-seeking responders

Lines 540-548: the authors should also include the mean probability of punishment for the information-seeking responders.

We have added the following statement (now lines 624-625): “The mean probability of punishment by information-seeking responders was 38.7% (SE = 0.023, $t(1001) = 16.580$, $p < 0.001$).” Thank you!

12) Figure 5 Adjustments

Figure 5. Similar comments as those provided for Figure 4.

We have changed the Y-Axis label to “Punishment (costly rejection)”, as in Fig. 4, SI-1, and SI-3 (see also responses to points 5 and 6 above). Thank you.

13) H10/H20 Reporting

Line 619: “but [we] could not reject”

We have added “we” to the initial line 619 (now line 709).

14) Reversal H3A Reporting

Line 618: Shouldn't be the other way around “We found a significantly higher probability of punishment by ignorant than by non-ignorant responders”? Punishment is defined as rejecting an unfair offer, isn't it?

Yes to both questions, thank you. We have reversed the phrasing in lines 708-709, accordingly.

15) Data Supporting Conclusion

Do the authors' results really support such an interpretation “We interpret these results as evidence for sorting Behavior in that individuals select their informational states based on their social preferences”? Can the authors better explain what they mean with “social preferences” here? Again, I don't think the data support this claim “Those willing to punish seek information about unfairness, while those unwilling to punish ignore it to reduce cognitive dissonance and lower emotional costs.” It seems to me that participants first decided whether to know or not and then they decided to whether accept or reject the offer, didn't they? If the above is correct, then the authors cannot have access to participants' preferences for willingness to punish before they decide to seek information. Therefore, their data doesn't really support their conclusion.

Thank you very much for your helpful feedback. Your feedback has contributed to greater clarity and readability and guided us to base our discussion section more directly on our results section. Specifically, we have significantly shortened our discussion to reduce

speculation on potential causes and matters that cannot be resolved. In particular, we have removed all content from lines 696 to 728 of our initial Stage 2 manuscript (incl. Figure 6). We have added a separate header for “Limitations” (now line 787) and we have inserted a reference to our a priori stated interpretation (now lines 753-755): “Rather, we interpret costly punishment as the reinforcement of a fairness norm in line with economic theories of fairness (for our pre-registered interpretations, see R10 and R20 in Table 1).” We have removed speculations on the reasons for not seeking information in both the abstract (now lines 28-29) and in the discussion (now lines 711–713): “We interpret these results as evidence for sorting behavior in that people who punish experienced unfairness actively seek information about unfair ultimatum offers, whereas those who decide not to punish tend to deliberately ignore such information.” Further, we have highlighted that future research is needed on the role of social preferences (now lines 758-759): “Yet, future research is needed to examine the role of SVO in the relationship between deliberate ignorance and UG punishment.” Finally, we have removed references to “social preferences” (initially present in line 622), as suggested.

Reviewer #2 Feedback:

0) Remarks to the Author

Overall, the authors have adhered to their registered study design and analysis plan. As a pre-defined outcome-neutral criterion, floor and ceiling effects in participants’ behaviour were checked. Introduction, rationale, and hypotheses are the same as in the registration. They have added some exploratory analyses to explain some of the failed rejections of stated null hypotheses, which are clearly marked as such. The exploratory analyses take up more room in presenting Results and their Discussion than the registered analyses. Moreover, the a priori stated interpretation of null effects for RQs 1 and 2 (“Reinforcement of a fairness norm based on altruistic punishment” and “Punishment in line with classic economics of information and economic theories of fairness”) receive comparatively little attention in the Discussion. However, the exploratory analyses were mostly done to illuminate possible mechanisms behind the hypothesized but not found effects, so this imbalance should probably not be interpreted too negatively. Yet I think, the paper might benefit from taking the initially stated interpretations more into account while discussing the findings. In sum, this is a well written paper providing interesting insights and a good discussion about a possible mechanism driving Behavior under uncertainty and inequality possibly dependent on (not measured) personality traits. Besides the issue of interpreting the null effects of RQs 1 and 2, I have only two minor comments to the authors.

Thank you very much for your positive assessment of our Stage 2 manuscript. We have significantly shortened our discussion section to reduce speculation on potential causes and matters that cannot be resolved. Details on all the content that we have removed from our discussion section can be found in response to point 15 by Reviewer 1 (see above). Further, we have re-stated our a priori interpretation to balance our exploratory and confirmatory findings (now lines 753-755): “Rather, we interpret costly punishment as the reinforcement of a fairness norm in line with economic theories of fairness (for our pre-registered

interpretations, see R10 and R20 in Table 1).” Your feedback has greatly contributed to a more balanced discussion section. Thank you!

2) Statement Overall model fit

Please state the overall model fit analyses for the first two models addressing RQs 1 and 2, subsumed under the heading “Uncertainty and Punishment”. At the moment, these statistics are only given for RQ3.

We have added the following two specifications to the “Uncertainty and Punishment” section to state the overall model fit for linear probability models (1) and (2):

1. The overall fit of our linear probability model (1) was not significant ($SE = 0.40$, $R^2 = 0.001$, $Adj. R^2 < 0.001$, $F(1, 1368) = 1.613$, $p = 0.204$).
2. The overall fit of our linear probability model (2) was significant ($SE = 0.39$, $R^2 = 0.046$, $Adj. R^2 = 0.044$, $F(3, 1366) = 21.78$, $p < 0.001$). Inequality significantly predicted punishment ($\beta_{22} = 0.18$, $SE = 0.030$, $t(1366) = 6.044$, $p < 0.001$).

We have further reordered the reporting of our results after adding specifications for the overall model fit for our linear probability models (1) and (2). Thank you.

3) Statement Confidence Interval

In the section “Ignorance and Punishment” in the Results, you state a confidence interval for the proportion of participants who did not seek information. This information confused me. Why is there a standard error to calculate a CI with for a proportion you know directly from participants behaviour?

We have removed the CI from the proportion of information-seeking participants in the “Ignorance and Punishment” section. Thank you for bringing this to our attention.

Reviewer #3 Feedback:

0) Remarks to the Author

This registered study entitled “Avoiding Punishment by Ignoring Unfairness: Responses to Uncertain Inequality in the Ultimatum Game” (COMMSPSYCHOL-23-0037C) investigated the effect of deliberate ignorance of uncertainty on the punishment (i.e., reject) in the Ultimatum Game (UG). Their results showed that the influence of uncertainty on the rejection rate was not significantly different from that of inequality. No interaction between uncertainty and inequality was observed. Instead, the authors found that participants who sought the information of distribution rejected the offer more than those who ignored the information. These results suggest that participants who had the will to punish sought the information, while those who did not have the will to punish avoided revealing the

information. Enough participants were recruited based on the results of pilot studies. The analyses, although simple, are sufficient to test the hypotheses. The results differ in some respects from what the authors expected, but the authors report honestly. The authors' interpretation of the results seems sound. Therefore, I have no concerns to be addressed.

Thank you very much for your positive evaluation of our Stage 2 manuscript and, once again, for your helpful feedback on our Stage 1 manuscript.